# Direct extraction of lithium from ores by electrochemical leaching

Hanrui Zhang [1], Ying Han[2], Jianwei Lai[1], Joseph Wolf [1], Zhen Lei[1], Yang Yang [2] & Feifei Shi [1] ✉

With the rapid increase in lithium consumption for electric vehicle applications, its price soared during the past decade. To secure a reliable and cost-effective supply chain, it is critical to unlock alternative lithium extraction resources beyond conventional brine. In this study, we develop an electrochemical method to directly leach lithium from α-phase spodumene. We find the $H_2O_2$ promoter can significantly reduce the leaching potential by facilitating the electron transfer and changing the reaction path. Upon leaching, β-phase spodumene shows a typical phase transformation to $HAlSi_2O_6$, while leached α-phase remains its original crystal phase with a lattice shrinkage. To demonstrate the scale-up potential of electrochemical leaching, we design a catalyst-modified high-throughput current collector for high loading of suspended spodumene, achieving a leaching current of 18 mA and a leaching efficiency of 92.2%. Electrochemical leaching will revolutionize traditional leaching and recycling processes by minimizing the environmental footprint and energy consumption.

As a key element in the production of lithium-ion batteries (LIBs), lithium has experienced a significant surge in demand, owing to the extensive utilization of electric vehicles[1,2]. The price of lithium carbonate, an essential precursor to synthesize commercial cathode materials, has increased five-fold in 2021[3]. There are multiple resources to harvest lithium, including brine, ores, and used LIBs[4–6]. Currently, over 70% of lithium-bearing chemicals are extracted from brines due to the low production cost[7]. Solar evaporation is utilized to separate $Li^+$ from other metal ions in brines, such as $Na^+$, $K^+$, and $Mg^{2+}$. However, this process has drawbacks, including a large footprint, lengthy extraction time, and soil pollution[8]. Furthermore, the nonuniform distribution of brine limits its production to regions like China and Chile[9]. It is essential to unlock more resources beyond brines to meet the increasing demand for lithium-based chemicals.

Lithium-bearing ores come in various forms, including spodumene, lepidolite, petalite, and Li-rich clays, uniformly distributed globally. Spodumene ($LiAlSi_2O_6$) is the primary ore used in lithium production due to its high concentration (~8 wt. % $Li_2O$)[10]. Spodumene can exist in α, β, and γ phases, where the α-phase is the natural state. γ-phase is a metastable phase when heating α phase between 700 °C and 900 °C[11,12]. By heating the α-phase above 1100 °C, the β-phase spodumene can be obtained with less density, i.e., less atom density per volume[11]. Traditionally, strong acids or bases are used to leach lithium under elevated temperatures[13]. For the strong acid leaching process, $Li^+$ is exchanged by proton[14], while $Li^+$ is released by breaking the Si-O bond in the base solution[15]. The main drawback of the acid leaching method is the substantial energy consumption during the high-temperature (1100 °C) calcination to form β-phase spodumene[16]. Basic leaching methods require lower operation temperatures, yet they need to separate impurity ions like $Al^{3+}$, $SiO_3^{2-}$, and $Na^+$ in the subsequent treatment[12,17]. Due to the limitations of existing chemical leaching methods, there is an urgent need for sustainable leaching techniques to reduce the high reaction barrier and minimize environmental impact.

The electrochemical leaching utilizes the electrical field to facilitate the metal ion dissolution[18], which has been applied on electronic wastes[19] and used LIBs[18,20]. Compared to chemical leaching methods, it surpasses the sole dependence on high temperature

[1]John and Willie Leone Family Department of Energy and Mineral Engineering, The Pennsylvania State University, University Park, PA 16802, USA. [2]Department of Engineering Science and Mechanics, The Pennsylvania State University, University Park, PA 16802, USA. ✉e-mail: feifeishi@psu.edu

and concentration of leaching agent to activate the reaction[21]. The sustainability of electrochemical leaching can be ensured by utilizing clean energy sources such as wind and solar for electricity generation[22]. However, the electrochemical reaction is limited by its heterogeneous nature, which usually leads to large overpotential, side reactions, and poor Faraday efficiency[23]. Soluble leaching promoters such as $H_2O_2$[24], $FeCl_3$[20], and $SO_2$[25] are added as the homogeneous agent to promote electron transfer kinetics. Among all the promoters, $H_2O_2$ is an ideal choice, as it can eliminate the ion-separation process. However, its tendency to decompose during transportation and storage raises safety concerns[19,26]. Additionally, poor mass transportation from traditional 2-dimensional current collectors hinders the electrochemical method from high throughput leaching[27]. To effectively address these limitations in electrochemical leaching, both electron and mass transportation must be improved for large-scale applications.

In this study, we developed an electrochemical leaching method to directly extract Li from α-phase spodumene using 0.5 M sulfuric acid at room temperature. We studied the effect of morphology, crystallographic structures, and surface chemistry on electrochemical leaching with scanning electron microscopy (SEM), transmission electron microscopy (TEM), X-ray diffraction (XRD), and X-ray photoelectron spectroscopy (XPS). $H_2O_2$ promoter is added to facilitate the electron transfer among the current collector, electrolyte, and spodumene. Its effectiveness is validated by thermodynamic calculation and linear sweep voltammetry (LSV). XRD and TEM results show that α-phase spodumene maintains its structure after electrochemical leaching, with only lattice shrinkage. Integrating atomic-resolution TEM imaging with the geometric phase analysis (GPA), the reaction frontier of electrochemical leaching was identified. AlOOH species is found on leached β-phase spodumene, while not observable on leached α-phase. The existence of intermediate product $O_2^{2-}$ is validated by in-situ Raman spectroscopy. We developed a 3-dimensional current collector with high surface area, and a proton-conducting binder, and modified it with Au nano-catalysts to achieve high-throughput long-term leaching. To evaluate our electrochemical leaching method, the Faradaic efficiency and leaching efficiency are monitored by the Chronoamperometry (CA) method and inductively coupled plasma atomic emission spectroscopy (ICP-AES). An overall 92.2% leaching efficiency was achieved, similar to traditional leaching methods. Compared to the traditional leaching method, the techno-economic assessment shows the cost of electrochemical leaching can be reduced by 35.6%, with 75.3% less $CO_2$ emission.

## Results

### Morphology, crystal structure, and surface chemistry of spodumene and their effects on electrochemical leaching

It has been found that the morphology and crystal structure of ores greatly impact the leaching efficiency[28]. We compared the spodumene morphology evolution after the phase transformation from α- to β-phase. As shown in Fig. 1a, pristine α-phase spodumene has a compact faceted structure where exposed planes are (110). Spodumene particles of β-phase (Fig. 1b) show pulverized and delamination structure after 1100 °C calcination. In Fig. 1c, X-ray diffraction (XRD) results reveal the α-phase spodumene has a typical monoclinic structure characterized by (020), (−221), and (310) planes, which are located at 21.17°, 30.63°, and 32.037°, respectively. In β-phase, only the (201) plane located at 25.45° shows a strong intensity, indicative of a tetragonal structure. The crystal structure of spodumene will influence the density of different phases. The density of the α-phase monoclinic structure is 3.178 g cm$^{-3}$, and that of the β-phase is only 2.375 g cm$^{-3}$. Supplementary Fig. 1 exhibits that the median particle size of α-phase is 51.0 μm. The median particle size of β-phase decreases to 31.1 μm after phase transformation. The difference in morphology and crystal structure explains the difficulty of directly extracting Li from α-phase

spodumene. Without phase transformation, the compact and dense structure hinders the Li$^+$-H$^+$ exchange during the acid-leaching process[14].

X-ray photoelectron spectroscopy (XPS) is a commonly used technology to probe the sample surface's elemental ratio and valence state. In the XPS survey (Supplementary Fig. 2), we found the Li:Al:Si stoichiometric ratio on the α-phase's surface is 0.85:0.83:2. Its elemental ratio in the bulk phase is 0.96:1.04:2, which is measured by inductively coupled plasma atomic emission spectroscopy (ICP-AES). All the ratios are normalized to Si. The higher Al and Si ratio comes from the associated minerals such as quartz. On the contrary, β-phase has a Li:Al:Si ratio of 0.70:0.88:2 on the surface and a ratio of 0.9:1.02:2 in the bulk phase. XPS depth profiling analysis confirms the Li concentration variation near the surface. As shown in Fig. 1d, the Li:Al ratio is around 1:1 in α-phase after 100 s sputtering. In contrast, the Li:Al of β-phase is 0.77:1. The deficiency of Li in both surface and bulk in β-phase spodumene might come from the evaporation during the high-temperature calcination, which has been widely reported in cathode materials and solid-state electrolytes for LIB[29].

To evaluate the valence state change on pristine α/β phase spodumene, Li 1 s, Al 2p, Si 2p, and O 1 s core level spectra are examined, as shown in Supplementary Fig. 3. The Li 1 s of α and β-phase are 55.7 and 55.8 eV, respectively, similar to $Li_2O$ (55.6)[30]. Meanwhile, Al 2p shifts from 74.6 eV (α-phase) to 74.2 eV (β-phase) after calcination. The Al 2p in α-phase is close to $Al(OH)_3$ (74.8 eV), and β-phase is similar to $Al_2O_3$ (74.1 eV)[12]. Si 2p spectra show the same binding energy (102.4 eV) for both phases[31]. O 1 s in α-phase is 531.8 eV, while 531.5 eV in β-phase. The knowledge of cation distribution and their valence state in pristine α/β -phase spodumene will help us understand the electrochemical leaching mechanism.

Linear sweep voltammetry (LSV) can determine the onset potentials of electrochemical reactions. Here, we used a 3-electrode cell with 0.5 M $H_2SO_4$ electrolyte, where α/β phase spodumene powders coated on a graphite rod are working electrodes. The graphite rod is the counter electrode, and a saturated calomel electrode (SCE) is the reference electrode. More details are provided in Method and Supplementary Fig. 4. Figure 1f shows that the β-phase has an oxidation peak at 1.09 V vs. SCE with a current of 0.83 mA, indicating Li is leached out. For α-phase, there is a very small leaching peak (0.55 mA) at 1.22 V, indicating the difficulty of directly extracting Li from α-phase. To check if Li is sufficiently extracted during the LSV, we perform subsequent cyclic voltammetry (CV) for three cycles in Supplementary Fig. 5. There is no leaching peak for either phase in the following cycles. Compared to the traditional acid leaching method, which solely relies on the Li$^+$-H$^+$ exchange, the electrical field can directly extract Li out of the spodumene lattice, analogous to deintercalation in classical Li-ion battery cathode materials[32]. In electrochemical leaching, the electric field can directly pull Li$^+$ from α-phase. However, electrolyte decomposition reactions, like oxygen evolution for aqueous electrolyte, compete with Li extraction reaction, as shown by the black curve in Fig. 1f. A large amount of electron leakage into the electrolyte will cause low Faradaic efficiency during the electrochemical leaching. Hence, lowering the electrochemical leaching potential is the key to efficiently extracting Li from α-phase spodumene.

### Thermodynamics of electrochemical leaching

To understand the origin of high leaching potential on α-phase spodumene, the Gibbs free energy change of leaching reactions are calculated, as reactions (1) and (2). The Gibbs free energy change ($ΔG$) determines whether a reaction is favorable or spontaneous, which can be connected to the reaction potential ($E_{rxn}$) of an electrochemical reaction[33]. As shown in Fig. 2a, the $ΔG$ of leaching in α-phase spodumene is 405.81 kJ mol$^{-1}$ at 298 K, and that of β-phase is 365.55 kJ mol$^{-1}$. This explains the higher leaching potential on α-phase, compared to β-phase in Fig. 1e. $ΔG$ decreases as temperature increases, showing that the

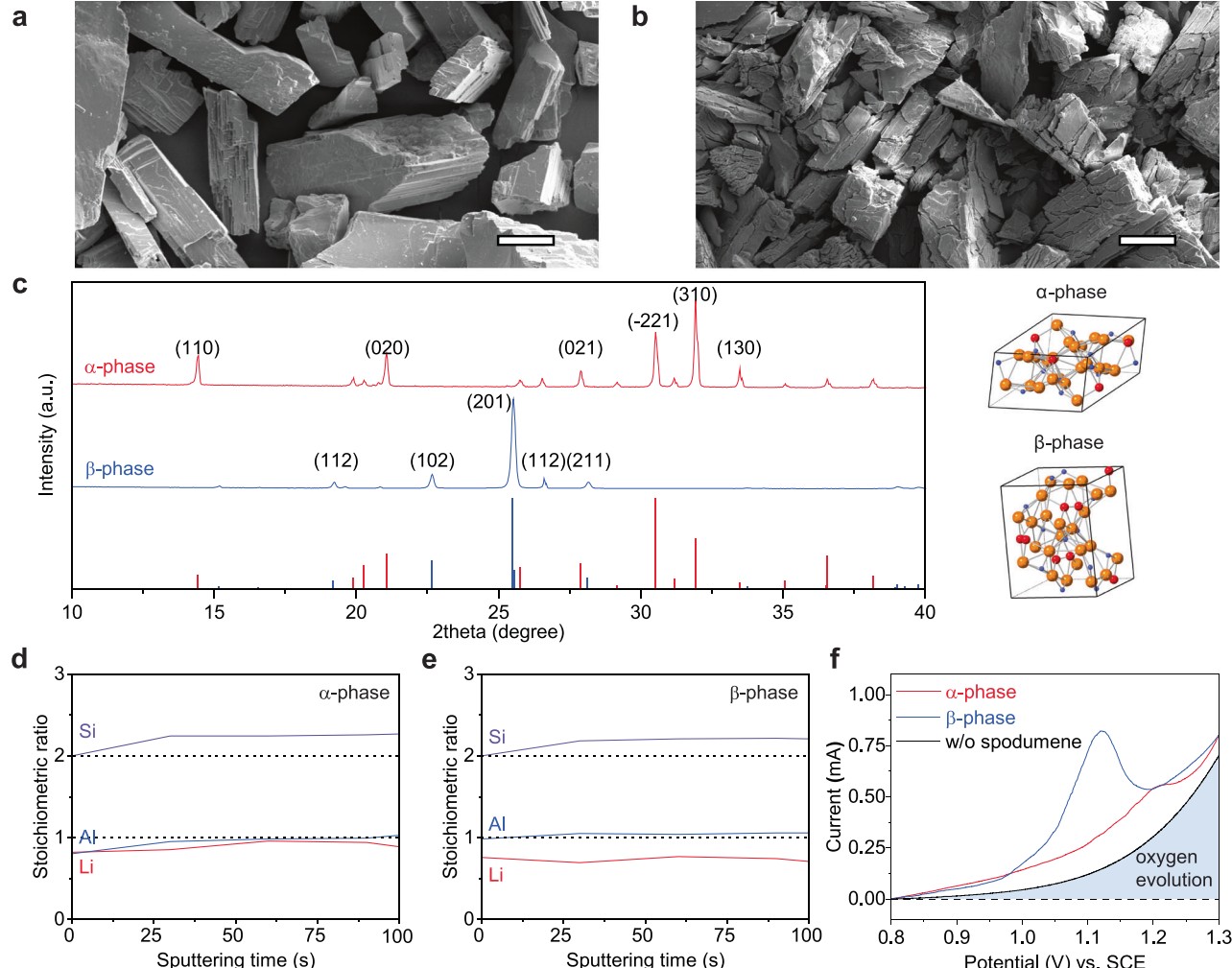

**Fig. 1 | Morphology, crystal structure, and surface chemistry of spodumene and their effect on electrochemical leaching.** Morphology of **a** α-phase spodumene and **b** β-phase spodumene. **c** X-ray diffraction (XRD) patterns of α and β-phase spodumene. Drop lines: PDF #98-000-0408 (red) and PDF #98-001-3106 (blue), and crystallographic structure of α and β-phase spodumene (red atoms−Li, orange atoms−O, blue atoms−Al or Si). Depth-profiling X-ray photoelectron spectroscopy (XPS) analysis of **d** α-phase and **e** β-phase spodumene. The Ar-ion sputtering rate is calibrated as 7.4 nm min⁻¹ by etching SiO₂ grown on the single crystal Si wafer. **f** electrochemical leaching test using linear sweep voltammetry (LSV) on α and β-phase spodumene and a graphite electrode with conductive carbon and Nafion binder in 0.5 M H₂SO₄ aqueous electrolyte at a scan rate of 0.5 mV s⁻¹. ~5 mg spodumene particles are coated on a graphite rod with conductive carbon and Nafion binder. The light blue region denotes the electrons consumed by oxygen evolution reactions. Scale bar: 20 μm.

leaching reaction is favored to happen at higher temperatures[34]. In the traditional acid leaching method, the leaching temperature is elevated to 250 °C for β-phase spodumene to improve the leaching efficiency.

Besides increasing the process temperature, adding a promoter to the electrochemical reaction can lower the reaction barrier. The promoter acts as an electron transfer agent, facilitating the electron transfer between the active materials and the leaching electrolyte[35]. Here, H₂O₂ was chosen as the promoter. After adding H₂O₂, both reactions (3) and (4) show a dramatic ΔG decrease. Assuming the leaching products are the same as that in reactions (1) and (2), the ΔG of reaction (3) is 53.52 kJ mol⁻¹, while that of (4) shifts to 6.08 kJ mol⁻¹ at 298 K. In Fig. 2b, we study the H₂O₂ effect on the electrochemical leaching reactions. With the addition of 0.5 wt.% H₂O₂, the leaching potential on α-phase spodumene is ~ 0.95 V vs. SCE. By shifting the leaching potential to lower voltage, O₂ evolution can be greatly minimized and further improve Faradaic efficiency. Compared to the electrochemical leaching without H₂O₂ in Fig. 1f, the leaching current in α-phase spodumene increases from 0.10 mA−1.96 mA at 0.95 V. In Fig. 2c, we vary the concentration of H₂O₂ from zero to 1 wt.% and identify the critical concentration to be 0.1 wt.%. Above this

concentration, electrochemical leaching of α-phase spodumene is activated. Higher concentrations will lead to more decomposition of H₂O₂ and O₂ evolution reactions. Hence, it's crucial to optimize the promoter concentration during the electrochemical leaching process.

$$2LiAlSi_2O_6(\alpha) + 2H_2O + 2H^+ \rightarrow 2HAlSi_2O_6(\alpha) + O_2 \uparrow + 2H_2 \uparrow + 2Li^+ \tag{1}$$

$$2LiAlSi_2O_6(\beta) + 2H_2O + 2H^+ \rightarrow 2HAlSi_2O_6(\beta) + O_2 \uparrow + 2H_2 \uparrow + 2Li^+ \tag{2}$$

$$2LiAlSi_2O_6(\alpha) + H_2O_2 + 2H^+ \rightarrow 2HAlSi_2O_6(\alpha) + O_2 \uparrow + H_2 \uparrow + 2Li^+ \tag{3}$$

$$2LiAlSi_2O_6(\beta) + H_2O_2 + 2H^+ \rightarrow 2HAlSi_2O_6(\beta) + O_2 \uparrow + H_2 \uparrow + 2Li^+ \tag{4}$$

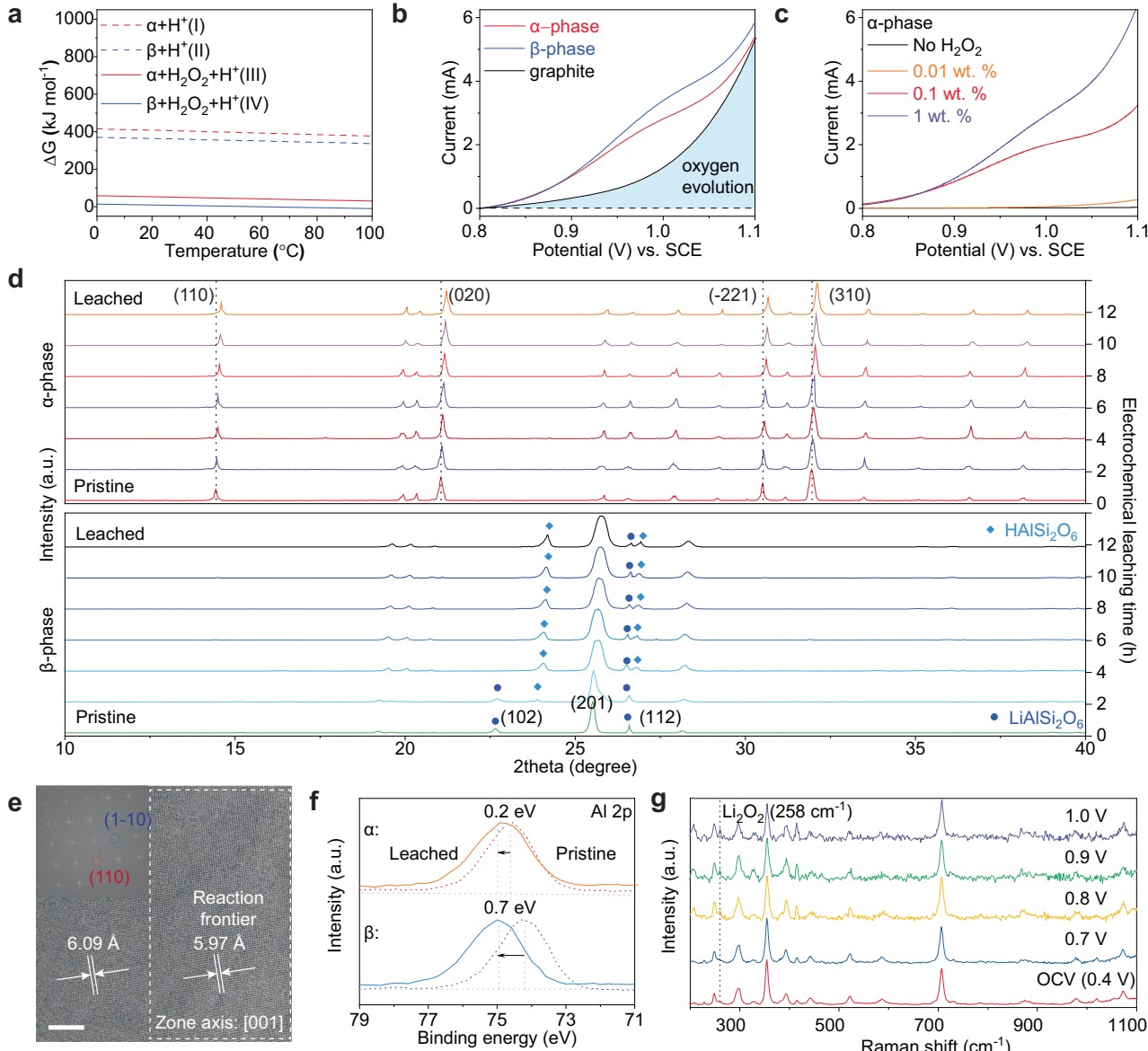

**Fig. 2 | Thermodynamic and kinetics study for electrochemical leaching with $H_2O_2$ promoter. a** Gibbs free energy change prediction of spodumene leaching reactions with/without $H_2O_2$ calculated by HSC chemistry software (Version 10.1). **b** Electrochemical leaching test with linear sweep voltammetry (LSV) on $\alpha$ and $\beta$-phase spodumene in 0.5 M $H_2SO_4$ aqueous electrolyte with 0.5 wt.% $H_2O_2$ as a promoter at a scan rate of 0.5 mV s$^{-1}$. **c** Electrochemical leaching performance with LSV using various $H_2O_2$ concentrations in 0.5 M $H_2SO_4$ aqueous electrolyte at a scan rate of 0.5 mV s$^{-1}$. - 5 mg spodumene particles are coated on a graphite rod with conductive carbon and Nafion binder **d** X-ray diffraction (XRD) patterns of pristine and leached spodumene as a function of electrochemical leaching time. **e** Lattice structure of $\alpha$-phase spodumene imaged by high-resolution transmission electron microscope (HRTEM). Insets: corresponding FFT pattern. **f** The X-ray photoelectron spectroscopy (XPS) spectra centered on Al 2p before and after leaching. **g** In-situ Raman spectra of $\alpha$-phase spodumene. The counter electrode is silver wire, and the voltage is converted to voltage vs. SCE. Scale bar: 10 nm.

## Electrochemical leaching mechanism with $H_2O_2$

It has been reported that the traditional acid leaching mechanism on $\beta$-phase spodumene is based on the H$^+$-Li$^+$ exchange, which highly relies on the acid concentration and temperature[28]. To understand the role of $H_2O_2$, we characterize the crystal structure evolution during the electrochemical leaching of $\alpha$ and $\beta$ phase, as shown in Fig. 2d. After holding the leaching potential at 0.95 V vs. SCE for 12 h, the (110), (020), (310) peaks in $\alpha$-phase positively shifted to 14.631°, 20.134°, 32.265°, respectively. The positive shift of these peaks is related to the Li extraction degree. However, the phase of HAlSi$_2$O$_6$ did not show up on $\alpha$-phase, indicating a different leaching pathway with $\beta$-phase. In $\beta$-phase, the (102) and (112) peaks of LiAlSi$_2$O$_6$ split into two peaks at 24.122° and 26.736° from HAlSi$_2$O$_6$, indicating ion exchange between

Li$^+$ and H$^+$[36]. Both phases are further leached for 24 h in Supplementary Fig. 7. The results show that Li can be completely extracted after 12 h. All the lattice constants that change during electrochemical leaching are summarized in Supplementary Table 3 and 4. The Li$^+$-H$^+$ exchange in $\beta$-phase is similar to that in the traditional acid leaching, while the direct extraction of Li from $\alpha$-phase spodumene does not follow the same path.

As a complementary tool of XRD, TEM can provide local crystal structure information during leaching. Here, we prepare the $\alpha$-phase TEM sample via the focused ion beam (FIB) lift-out, which is thinned down to ~100 nm. More FIB experimental details are provided in the Method section and Supplementary Fig. 8. We resolve the atomic structure of pristine $\alpha$-phase spodumene, as shown in Fig. S9a. The

pristine (110) plane has a planer spacing of 6.09 Å, consistent with the XRD result. After the leaching, the interplanar spacing shrinks to 5.96 Å, as shown in Supplementary Fig. 9b, indicating a 2.13% reduction of plane spacing after electrochemical leaching. The electron diffraction patterns shown in Supplementary Fig. 10 demonstrate the α-phase remains a monoclinic structure before and after the leaching. A half-leached sample was characterized by TEM to identify the reaction frontier. As shown in Fig. 2e, the TEM image and corresponding fast Fourier transform (FFT) pattern confirm that the lattice of the α-phase spodumene shrinks after electrochemical leaching. Localized strain or plane-spacing change can be measured from HRTEM images using the geometric phase analysis (GPA) algorithm as reported[37]. In Supplementary Fig. 11, the GPA result of $\varepsilon_{xx}$ is presented, allowing for identifying pristine, leached areas and the intermediate reaction frontier. The GPA map shows that (110) interplanar distance shrinks ~ 2% after the electrochemical leaching, consistent with the XRD and TEM results. Understanding the preferred reaction pathway can help us further enhance the effectiveness of electrochemical leaching, such as exposing the preferential planes by chemical leaching in advance.

We used XPS to check the chemical species' evolution during electrochemical leaching. After leaching, Al 2p of β-phase shifts to 75.1 eV in Fig. 2f. This is close to typical AlOOH (75.2 eV) in aluminum ore. Al 2p of α-phase slightly shifts from 74.6 to 74.9 eV. In Supplementary Fig. 12d, O 1s of α-phase shifts positively from 531.8 to 532.6 eV. Comparing the results in Supplementary Fig. 12b and d, we found that the Al-O bond in the leached α-phase is neither $Al(OH)_3$ nor AlOOH. Besides the formation of O-H functional group, the higher valence state of oxygen, such as $O_2^{2-}$ and $O_2^-$, will also cause a positive shift of binding energy[38]. The formation of $O_2^{2-}$ intermediate species has been widely discovered during the deintercalation process of $Li^+$ in cathode materials of Li-ion batteries ($LiNi_xCo_yMn_zO_2$, $LiM_2O_4$)[32,39]. To confirm that only $Li^+$ is extracted, we test the leaching residue with both Energy-dispersive X-ray spectroscopy (EDS) and ICP-AES. As shown in Supplementary Fig. 13 and Supplementary Table 2, the Al to Si ratio is close to 1:2. Hence, the role of the $H_2O_2$ promoter is to facilitate the oxidation of $O^{2-}$ in the spodumene lattice to higher valence species, which ultimately dissolves lithium from α-phase at a mild condition.

$O_2^{2-}$ is a meta-stable species, and it is challenging to observe via ex-situ characterization tools directly[39,40], where the electron energy loss spectroscopy (EELS) and Raman are mainly used to validate it. To understand the mechanism of $O_2^{2-}$, we combined cryo-STEM coupled with EELS and in-situ Raman spectroscopy. The cryo-STEM/EELS sample preparation follows the standard lift-out protocols, and a cryo-holder was employed and operated at cryo-temperature to avoid electron beam-induced damage, as shown in Supplementary Fig. 14. The EELS spectra are shown in Supplementary Fig. 15, indicating a difference in oxygen ratio before/after leaching, but the position of O doesn't show an apparent shift. It could come from our experiment conducted under the aqueous system and an ambient environment, the active $O_2^{2-}$ quickly reacted with the $H^+$ or $H_2O$ and formed stable species such as $O^{2-}$. Supplementary Fig. 16a and b exhibit the ex-situ Raman spectra of both phases before and after the electrochemical leaching, where the α-phase shows a similar pattern before/after leaching, consistent with the EELS results. Hence, the ex-situ test cannot trace the $O_2^{2-}$, and conducting an in-situ Raman test is necessary. A home-built three-electrode cell uses a Ag wire counter electrode and a Ag wire reference electrode. The working electrode is prepared by coating the Au-coated stainless-steel disk with α-spodumene/Nafion binder at a ratio of 9:1. We first increase the leaching voltage and monitor the in-situ spectra of the spodumene. The voltage interval is 0.1 V, and each voltage is held for 2 min to reach equilibrium. As shown in Fig. 2g, during the electrochemical leaching, as the leaching voltage increases to 0.8 V vs. SCE, it starts to show the feature of solid-state $Li_2O_2$ (258 cm$^{-1}$) from spodumene[41], demonstrating that

$O_2^{2-}$ is an intermediate product during the electrochemical leaching. The signal starts to weaken when voltage is above 1 V vs. SCE, mostly due to the oxygen evolution reactions. We also examine the in-situ Raman with different leaching times, present in Supplementary Fig. 16c. As the leaching time increases to 6 mins, we can observe the existence of the $O_2^{2-}$. To sum up, the $O_2^{2-}$ is a reaction intermediate during the electrochemical leaching of the α-phase spodumene, and the addition of $H_2O_2$ facilitates the reactions by increasing the concentration of $O_2^{2-}$.

## High throughput current collector design

Increasing feedstock loading per batch and total current during electrochemical leaching is essential to make this electrochemical leaching practical for large-scale applications. Suspended electrodes can dramatically increase the feedstock by suspending electroactive materials in a liquid electrolyte[42]. The suspended electrode significantly enhances scalability by promoting efficient utilization of active materials, transitioning from a heterogeneous to a homogeneous reaction regime with the assistance of promoters/mediators[43]. Given that α-phase spodumene represents its natural state and the phase transition to β-phase is energy-intensive, our research focuses on the α-phase to ensure that the leaching process is environmentally sustainable. We disperse 1 g spodumene particles in 50 ml 0.5 M $H_2SO_4$ with the 0.5 wt.% $H_2O_2$ promoter and constantly stirred at 500 rpm, as seen in Supplementary Fig. 17. In this case, we can process at least 50 times more spodumene than coating it onto the graphite rod, greatly improving the scalability of this system. We synthesized a 3-dimensional current collector using graphene oxide (GO) aerogel with a carbon felt (CF) framework to increase the current. GO has a high surface area, which accounts for more active sites for electrochemical reactions. It is a superior electrical conductor and chemically stable, allowing it to be used in extreme pH conditions. However, the GO foam is rigid and easy to break, so carbon felt is introduced as a framework to offer flexibility for the whole structure. As shown in Fig. 3a, GO formed a secondary structure inside the porous carbon felt after freeze drying, significantly enlarging the surface area. However, the adhesion between the GO flasks and carbon felt is weak, and the GO flasks easily delaminate due to flow turbulence. To stabilize the current collector, organic binders are added to the GO solutions before the freeze-drying. Three binders, including Nafion, polyvinylidene fluoride (PVDF), and carboxymethyl cellulose (CMC)/ styrene-butadiene rubber (SBR), are tested to optimize the electrochemical performance. As shown in Fig. 3b, the Nafion binder shows the highest current (2.01 mA) and lowest reaction potential (0.95 V vs. SCE), attributed to its excellent proton-conducting capability[44]. As a comparison, the current of the electrode using PVDF and CMC/SBR binders is only 1.82 and 1.76 mA, and the leaching potential is 1.02 V vs. SCE. Thus, the Nafion is chosen as the binder to reinforce the current collector, which will be noted GO-CF in the following section.

$H_2O_2$ promoter will be consumed over time as a reactant and decompose on the electrode as a side reaction during the leaching process. To maintain the leaching reaction, we have to maintain the critical concentration of 0.1 wt.%. In-situ electrosynthesis $H_2O_2$ with catalysts has been well studied in literature[45]. Since the electrochemical leaching potential for spodumene is relatively high (>0.9 V vs. SCE), we can only utilize noble catalysts, such as gold (Au)[46]. Here, we use the chronoamperometry method to load the Au catalyst on the 3D current collector. The loading is around 0.53 mg cm$^{-3}$. After the electrodeposition, the current collector is freeze-dried again to maintain its large surface area, and the current collector with the Au loading is noted as Au-GO-CF. The SEM image of Au nanoparticles is shown in Fig. 3c, where the nanoparticles are homogeneously distributed on the GO flask. Nano-sized Au particles are also characterized by HRTEM (Fig. 3d and Supplementary Fig. 20), showing a crystalline structure with an average particle size < 5 nm. During the electrochemical

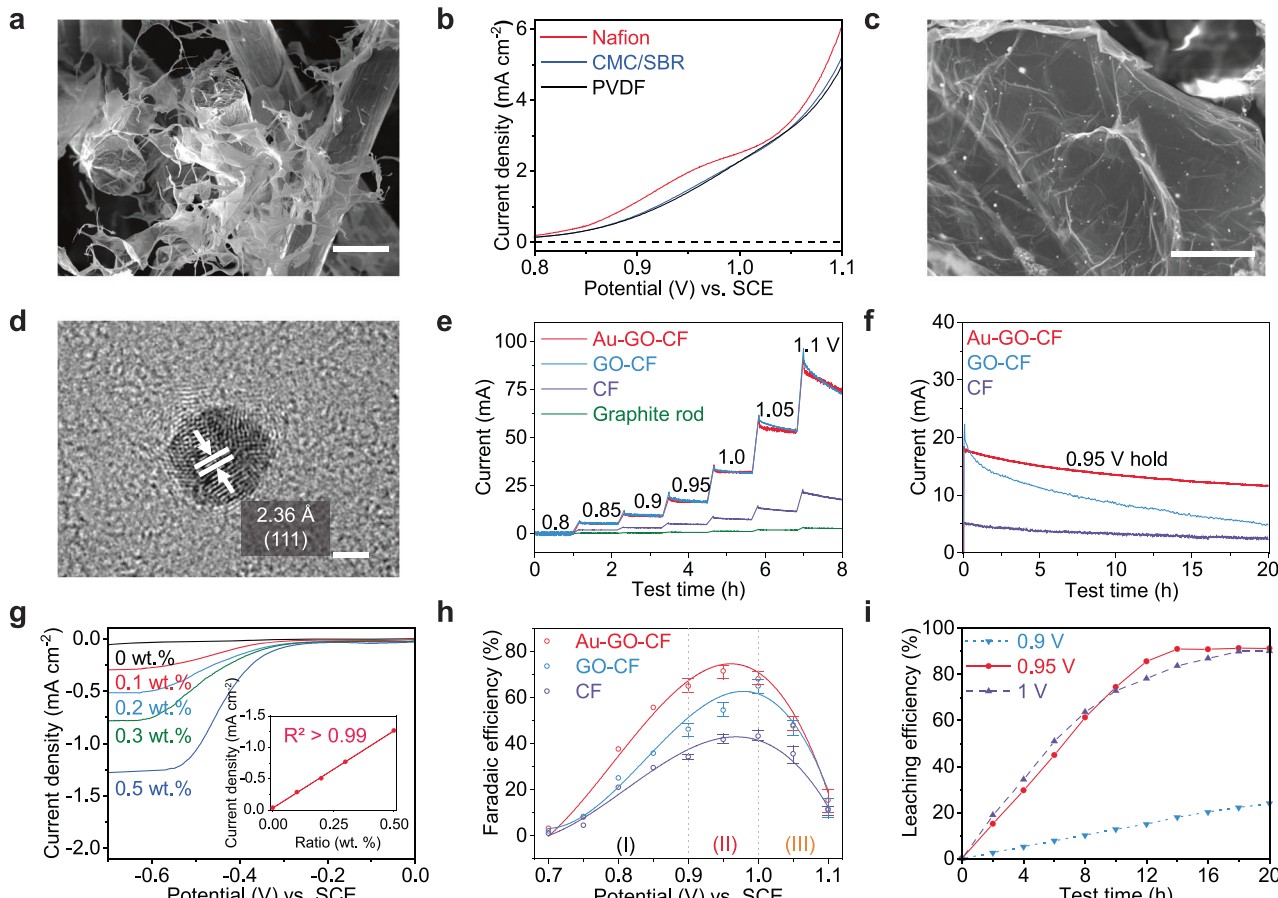

**Fig. 3 | High throughput current collector design and leaching conditions optimization. a** The morphology of the graphene oxide (GO)-carbon felt (CF) current collector characterized by SEM. **b** Electrochemical leaching of suspended α-spodumene on the 3D current collector (GO-CF) prepared with various binders (Nafion, PVDF, CMC/SBR). **c** SEM image of Au nanoparticles on Au-GO-CF. **d** high-resolution transmission electron microscope (HRTEM) image of a single Au particle. **e** Chronoamperometry test on different current collectors, voltage is controlled from 0.8 V to 1.1 V vs. SCE, and each voltage step is held for 1 h. **f** Stability test of electrochemical leaching on current collectors of CF, GO-CF, Au-GO-CF, leaching voltage is held at 0.95 V vs. SCE. **g** The in-situ detection of $H_2O_2$ concentration by microelectrode, reverse linear sweep voltammetry (LSV) curve is collected from 0 V to −0.7 V vs. SCE in 0.5 M $H_2SO_4$ with various $H_2O_2$ concentration, at a scan rate of 0.5 mV s$^{-1}$. inset: the linear fitting of the cathodic current density as a function of $H_2O_2$ concentration. **h** Faradaic efficiency of the electrochemical leaching reactions on different current collectors, voltage is held from 0.7 V to 1.1 V vs. SCE. The error bar is from 3 times repeat. **i** Leaching efficiency as a function of leaching potentials with Au-GO-CF current collector. For all scale-up electrochemical leaching with 3D current collectors, 1 g spodumene is suspended in the 0.5 M $H_2SO_4$ with 0.5 wt.% $H_2O_2$ electrolyte. Scale bars in SEM and TEM: **a** 100 μm. **c** 500 nm. **d** 2 nm.

leaching, the air is continuously purged into the electrolyte, and Au can catalyze the generation of $H_2O_2$. As shown in Fig. 3e, when leached at 1 V vs. SCE, the current densities of Au-GO-CF and GO-CF are 32 mA, while that of CF is only 7.9 mA. At other leaching potentials, Au-GO-CF and GO-CF also show 4−5 times higher currents than CF. It should be noted that the current is close but rapidly decays under high leaching potentials (>1V vs. SCE), indicating severe side reactions, such as the decomposition of $H_2O_2$. To improve the energy efficiency during the leaching, we must evaluate and identify these side reactions.

The stability of current collectors plays a crucial role in determining the lifespan and cost of the manufacturing process. To evaluate the stability of different current collectors, the potential is held at the leaching potential (0.95 V vs. SCE), as shown in Fig. 3f. The Au-GO-CF current collector shows the most stable performance with a retention of 65.26% after 20 h of hold test, primarily due to the in-situ formation of $H_2O_2$. On the other hand, the GO-CF current collector has a high initial current of 22.36 mA but quickly decays due to the rapid decomposition of the promoters and only obtains 22.84% of the initial current. The CF current collector is stable (retention of 47.25%) due to its low promoter consumption rate, but the current is only 5 mA at the beginning. This catalyst-modified Au-GO-CF current collector can

lower the cost of the manufacturing process as well as increase efficiency and productivity. It is also possible to perform as a closed-loop system without adding $H_2O_2$.

## Electrochemical leaching condition optimization

As discussed previously, $H_2O_2$ concentration near the electrode is crucial to initiate the reaction, so its concentration evolution must be carefully monitored and regulated. In this study, we use a glassy carbon microelectrode (34 μm diameter) to measure the $H_2O_2$ concentrations (details in Supplementary Note 1 and Supplementary Fig. 21). As shown in Fig. 3g, we first do the LSV scan from 0 V to -0.7 V vs. SCE in 0.5 M $H_2SO_4$ with various concentrations of $H_2O_2$, at the scan rate of 0.5 mV s$^{-1}$. In contrast, the linear scan for microelectrodes results in a diffusion limit current plateau (instead of a current peak) due to a steady-state response from the 3-D diffusion[47]. The phenomenon is analogous to a rotating ring-disk electrode (RRDE). The diffusion limit current comes from the electrochemical reduction of $H_2O_2$, and it reaches equilibrium when the potential is <-0.6 V vs. SCE. The inset of Fig. 3g presents the linear fitting of the diffusion limit current at -0.65 V as a function of $H_2O_2$ concentration, and the coefficient of determination ($R^2$) is > 0.99, showing a good

correlation between the $H_2O_2$ concentration and cathodic diffusion limit current density[48,49]. The microelectrode determines the $H_2O_2$ concentration in Supplementary Fig. 21c, and it can be seen the leaching current is directly related to the $H_2O_2$ concentration. As the $H_2O_2$ concentration decreases in Supplementary Fig. 21c, the leaching current in Fig. 3f also decreases. The Au-GO-CF current collector has the best current retention because it can retain the highest $H_2O_2$ concentration of 0.30 wt.% after 20 h of leaching, due to its ability to in-situ form $H_2O_2$.

Faradaic efficiency (FE) is a factor to evaluate the effectiveness of the electrochemical reaction by comparing the amount of substance produced to the number of electrons passing through the cell. Faradaic efficiency (FE) is usually expressed as a percentage, and it is defined as the amount of collected product to the theoretical amount of product that can be produced from the total charge passed[50]. Figure S22 shows the schematic of calculation and FE under different voltages. Figure 3h summarizes the FE of different current collectors at various leaching potentials and corresponding $H_2O_2$ concentrations. Compared to the pristine CF, Au-GO-CF current collectors significantly enhance the Faradaic efficiency of the reactions, with the highest FE of 71.5% at 0.95 V vs. SCE. The highest Faradaic efficiency occurred between 0.9 V–1.0 V vs. SCE, as lower potentials (region I, <0.9 V vs. SCE) are not sufficient to trigger lithium leaching reactions, while higher potentials (region III, >1 V) lead to more side reactions (oxygen evolution, decomposition of $H_2O_2$), and localized $H_2O_2$ concentration suddenly dropped after 1 V, as shown in Supplementary Fig. 22d. In contrast, the highest leaching efficiency of GO-CF and CF is 64.9% and 43.2% (at 1 V vs. SCE), respectively. The optimal leaching potential of Au-GO-CF is lower than the other two current collectors, which can be attributed to the higher localized promoter concentration due to the in-situ $H_2O_2$ formation, as confirmed by the microelectrode test. The overall FE shows a volcano shape, and the maximum FE is reached in region II (0.9–1 V), where the leaching reaction is activated, but the side reactions aren't initiated.

The leaching efficiency indicates the effectiveness of a leaching reaction, which is calculated as the ratio between extracted ions over the total amount of ions in raw materials[51]. Leaching efficiency is calculated as the ratio of the lithium ions in leachant (the electrolyte in this work) to the total Li content from pristine α-phase spodumene. It's derived based on the ICP-AES result, which is not influenced by the electrochemical signal. Leaching efficiency from rocks and minerals depends on several factors, including the strength of the acid, the type of mineral being leached, and the temperature and pH of the leaching solution. ICP-AES is used to examine the Li content in the leachant to determine the leaching efficiency and speed at various potentials. A leaching efficiency of 92.2% can be achieved, as exhibited in Fig. 3i. The Au-GO-CF current collectors are used, and the details can be seen in Supplementary Table 5. No lithium is leached out in the same solution for 24 h without the applied voltage. It can be found that 0.95 V vs. SCE is the optimized potential for achieving the highest leaching efficiency, where we reached 92.2% after 12-h electrochemical leaching. At the lower leaching potential (0.9 V vs. SCE), only 24.3% leaching efficiency is achieved after 20 h. The leaching rate is constant, showing that the reaction proceeds slowly. At the higher potential region (1.0 V vs. SCE), the leaching speed is fast at the beginning (from 0 to 8 h) but gradually slows down and doesn't reach the maximum leaching efficiency until 20 h, resulting from more side reactions such as the OER and the decomposition of the $H_2O_2$. The XRD patterns at various leaching potentials are listed in Supplementary Fig. 23, and the lattice shifts show the spodumene is fully leached at 0.95 V and 1 V vs. SCE but not fully leached at 0.9 V vs. SCE, indicating 0.95 V vs. SCE is the optimal leaching potential. 0.95 V vs. SCE is selected as the optimized leaching potential, carefully balancing the conditions that promote high Faradaic efficiency and leaching efficiency with minimal side reactions. At this leaching potential, the Faradaic efficiency reaches its peak of 71.5%

due to the efficient utilization of electrons for the leaching reaction rather than side reactions. Additionally, the leaching efficiency also achieves the highest value of 92.2%, as higher potentials led to faster initial reactions but more side reactions, ultimately lowering the leaching efficiency. This leaching potential can make full use of the energy while achieving the highest leaching degree.

## Discussion

Figure 4a summarizes the working mechanism of our synthesized high-throughput, catalyst-modified current collector for electrochemical leaching on suspended spodumene particles. This 3-dimensional current collector with GO and CF framework enlarges the unit process capability, and the proton-conducting Nafion binder enables the fast transport of $H^+$. With the continuous Air/$O_2$ input, Au catalysts will produce $H_2O_2$ locally, with microelectrode monitoring its concentration online. In addition to the batch mode, we also test the flow mode electrochemical leaching, as shown in Supplementary Fig. 24 and Supplementary Movie 1. Supplementary Fig. 25 shows the comparison of flowcharts to extract lithium from brines and spodumene. Figure 4b shows the state-of-the-art Li extraction from spodumene ores, compared with this work[15,17,28,52,53]. In traditional leaching, high temperature and chemical agent concentration are the driving forces for extracting Li[13]. High pressure is also a strategy to activate the leaching process and lower the leaching temperature[15,53]. Among all the leaching methods, our electrochemical leaching method requires less processing time, chemical agent usage, and energy consumption. We further estimate the cost and $CO_2$ emission for traditional and electrochemical leaching of our work, as shown in Fig. 4c, d. By minimizing the usage of the leaching chemicals, the cost of electrochemical leaching is reduced by 35.6%. By lowering the leaching processing temperature to room temperature, electrochemical leaching can reduce $CO_2$ emission by 75.3%, assuming the electricity is generated by fossil fuel-based power plants. More details of the techno-economic assessment and gas emission evaluations are provided in Supplementary Table 6 and 7, and Supplementary Note 2. As the starting point of the lithium battery supply chain, lithium extraction is critical to the success of the entire lithium battery industry. Compared to traditional leaching methods, electrochemical leaching doesn't rely on elevated temperature, high pressure, or high concentration of leaching agent, electrical field can effectively drive the leaching equilibrium. The sustainability of electrochemical leaching can be further ensured by utilizing clean energy sources such as wind and solar.

In this study, we successfully extracted lithium in α-phase spodumene at room temperature with dilute acid (0.5 M $H_2SO_4$). Though β-phase spodumene is most commonly used for Li extraction, a portion of lithium is lost during the high-temperature calcination. We found that adding the promoter ($H_2O_2$) will enable the direct leaching process on the α-phase by altering the $Li^+$-$H^+$ exchange pathway. For leached α-phase spodumene, both XRD and TEM show lattice shrinkage without the presence of new phases. Coupling with the GPA method, the reaction frontier is identified between the (110) planes. The $H_2O_2$ activates the leaching reaction by oxidizing the Al-O bond in the spodumene framework. In-situ Raman results demonstrate that $O_2^{2-}$ is an intermediate product during the reactions. For large-scale leaching, we designed 3-dimensional current collectors with large surface area, proton-conducting network, and Au catalysts to promote electron and proton transfer and generate $H_2O_2$ in-situ. We successfully leached lithium from suspended spodumene particles, providing great potential for a high-loading process up to 5 grams. After 12 h, the leaching efficiency achieved 92.2%, which is comparable to traditional leaching methods, but the energy consumption was reduced by 90%. As the demand for sustainable mining and recycling processes grows, electrochemical leaching will be widely adopted for critical element extraction.

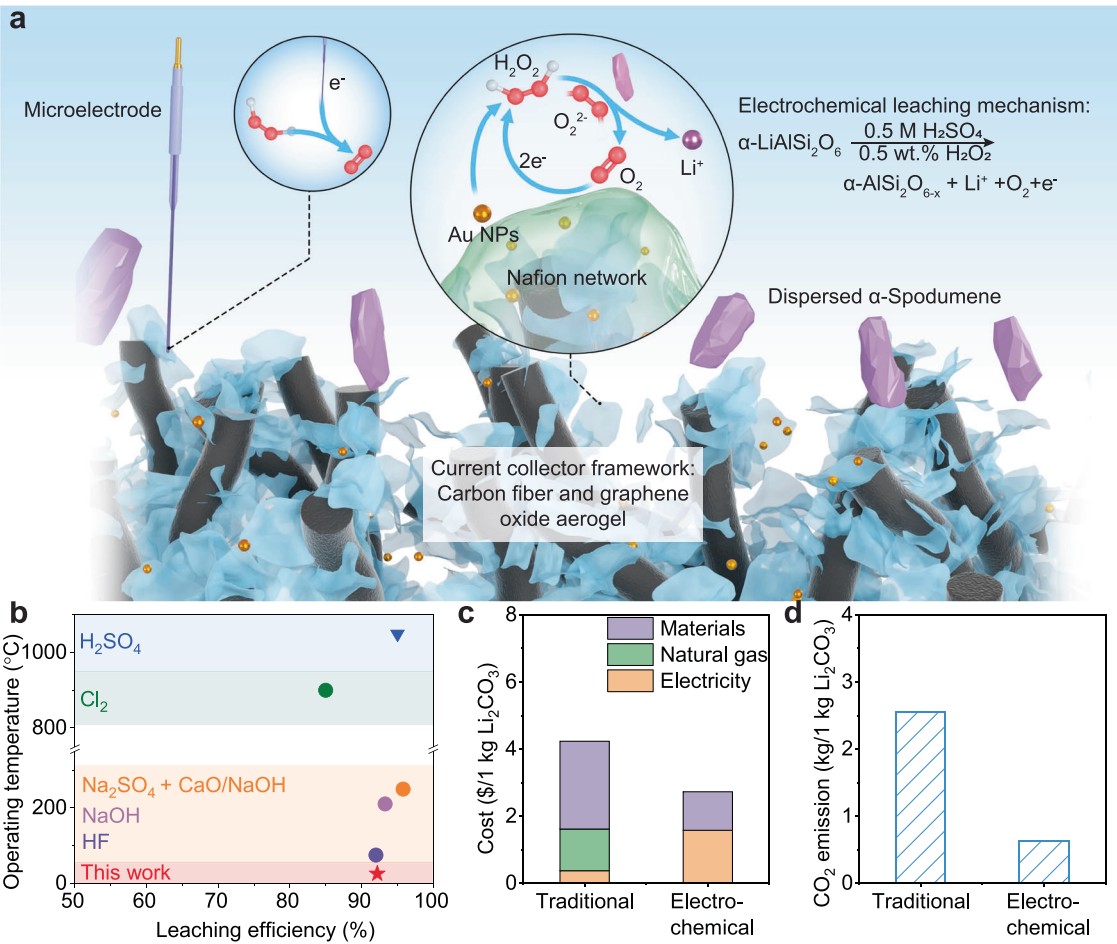

**Fig. 4 | Working mechanism of electrochemical leaching and state-of-art for Li extraction from spodumene feedstocks. a** The schematic of the working mechanism of our synthesized high-throughput, catalyst-modified current collector. Microelectrode: online monitor $H_2O_2$ promoter concentration near-current collector. CF and GO framework: increase surface area and conduct electrons. Nafion network: fast proton conducting path. Au catalysts: catalyze the reaction of in-situ $H_2O_2$ formation. Spodumene particles are dispersed in the electrolyte as the slurry electrode. **b** comparison of operating temperature and leaching efficiency for state-of-the-art leaching methods and this work[15,17,28,52,53]. Besides high

temperature, the concentration of chemical agents and the pressure of the reaction will influence the leaching efficiency. For the method with $Na_2SO_4$ and NaOH high-pressure treatment is used to improve efficiency. **c** Techno-economic assessment of traditional and electrochemical leaching method. Labor, buildings, and infrastructure are not included in the cost estimation. **d** $CO_2$ emission assessment of traditional chemical and electrochemical leaching methods. For (**c**) and (**d**) all the electricity is assumed from the fossil fuel-based power plants. The emission data for electricity was retrieved from the EIA emission data for electric plants in the United States (2021).

## Methods

### Materials

The α-phase spodumene was collected from the Carolina Tin-Spodumene Belt near Kings Mountain, NC. The β-phase spodumene was obtained by calcinating α-phase spodumene at 1100 °C for 12 h (air atmosphere). Carbon black, PVDF (polyvinylidene fluoride), and CMC (carboxymethyl cellulose)/SBR (styrene-butadiene rubber) were purchased from Xiamen TMAX Battery Equipment Limited and used as received. Nafion (5 wt% in IPA/water, Alfa Aesar), NMP (N-Methyl-2-pyrrolidone), HAuCl₄, $H_2SO_4$(98 wt.%), and $H_2O_2$ (30 wt.%) were purchased from Sigma-Aldrich and used as received. Triple-distilled deionized water was used for all aqueous electrolytes.

### Electrode preparation

The active materials (α-phase or β-phase spodumene) were mixed with conductive carbon and binder with a weight ratio of 6:3:1, dissolved in either water or NMP (based on the binder). The mixture was then sonicated for 2 h and coated onto a graphite rod. Al current collectors were coated by the doctor blade method. The mass loading

is ~ 5 mg cm⁻². The coated electrode was then dried at 100 °C under a vacuum to remove the solvent and moisture.

### Electrochemical test

The electrochemical tests were conducted using (Biologic) and PMC-1000 series (AMETEK, Inc.). The graphite rod coated with spodumene materials, or an as-designed current collector was used as the working electrode. Graphite rod (for small scale) and carbon paper (for large scale) were used as the counter electrode; the Saturated calomel electrode (SCE) serves as a reference electrode for the electrochemical leaching test in the aqueous electrolyte test. Li foil is used as both a counter and reference electrode for the aprotic electrolyte test. Linear sweep voltammetry (LSV) and cyclic voltammetry (CV) are conducted at the scan rate of 0.5 mV s⁻¹ in 0.5 M $H_2SO_4$ with various concentrations of $H_2O_2$ (0–1 wt.%). Electrochemical leaching is conducted by holding at 0.8–1.1 V vs. SCE. A home-built three-electrode cell is employed, using a Ag wire counter electrode, and a Ag wire reference electrode. The working electrode is prepared by coating the Au-coated stainless-steel disk with α-spodumene/Nafion binder (9:1).

## Current collector fabrication

The carbon felt was purchased and cleaned using water, ethanol, and acetone, subsequently via sonication. The cleaned carbon felt was then freeze-dried with $0.5\,g\,L^{-1}$ GO solution with 5% Nafion (w/w). The as-prepared electrode is noted as GO-CF current collector. The GO-CF electrode was placed in a 0.5 mM $HAuCl_4$ solution with 0.5 M $H_2SO_4$ as the supporting electrolyte. After applying a potential of -1 V vs. SCE for 10 s, gold nanoparticles were uniformly distributed onto GO and carbon fiber and freeze-dried again to maintain the open-porous structure. It is referred to as the Au-GO-CF current collector.

## In-situ monitor $H_2O_2$ concentration

A 34 μm carbon microelectrode (Kation Scientific) was used in this experiment as an electrochemical $H_2O_2$ concentration detector. The electrochemical apparatus contains four electrodes – two working electrodes (current collector and microelectrode), one counter electrode, and one reference electrode. The connection of electrical wires is similar to that of a rotating ring-disk electrode (RRDE). To synchronize the measurement of Faradaic efficiency with $H_2O_2$ concentration, an anodic leaching current was collected from the designed current collector (the 1st working electrode), while the cathodic current for $H_2O_2$ reduction was monitored on microelectrode (2nd working electrode). The microelectrode was placed 5 mm apart from the leaching current collector to measure the local promoter concentration. To correlate the cathodic current and concentration of $H_2O_2$, the microelectrode was tested in the electrolyte (0.5 M $H_2SO_4$) with various $H_2O_2$ concentrations (from 0 wt.% to 0.5 wt.%). Reverse scanned LSV curves are obtained from 0 V to -0.8 V vs. SCE. Below the $H_2O_2$ reduction potential (-0.65 V vs. SCE), $H_2O_2$ is continuously reduced[49]. Due to the 3-dimensional diffusion nature of microelectrode, the LSV curves show platform shape (e.g. limiting current)[47]. The calibration curve is developed by linear fitting the cathodic currents (at -0.65 V vs. SCE) with $H_2O_2$ concentration. With this established relationship, we can derive the unknown concentration of $H_2O_2$ near the current collector or any place in the leaching system.

## Characterization

Crystallographic characterization of the spodumene solid and the leaching residue was conducted through X-ray diffraction (XRD) analysis using Malvern Panalytical Empyrean I. The data was collected over a 2θ range of 10–70° with Cu Kα radiation at 40 kV and 30 mA. The morphology of the samples was studied by Verios (Thermo Fisher Scientific) scanning electron microscopy (SEM). For SEM analysis, each sample was coated with 5 nm thick iridium using a Leica Sputter Coater to mitigate the charging effect. The particle size analysis used Mastersizer 3000 (Malvern Panalytical) with 632 nm laser and 470 blue LED. The spodumene particles were dispersed with DI water. The lithium content of leachant was analyzed using Thermo iCAP 7400 Inductively Coupled Plasma–Atomic Emission Spectrometry (ICP-AES) at the Penn State Laboratory for Isotopes and Metals in the Environment (LIME). XPS measurements were conducted in a Physical Electronics VersaProbe II instrument equipped with a monochromatic Al kα x-ray source (hν = 1,486.6 eV) and a concentric hemispherical analyzer. The X-ray source generated a 200-μm diameter beam on the specimen. Charge neutralization was performed using both low-energy (< 5 eV) electrons and argon ions. The binding energy axis was calibrated using the carbon signal (C1s 294.8 eV) on the sample substrate. Quantification was done using instrumental relative sensitivity factors (RSFs) that account for the X-ray cross-section and inelastic mean free path of the electrons. The Ar-ion sputtering rate is calibrated as $7.4\,nm\,min^{-1}$ by etching $SiO_2$ grown on the single crystal Si wafer. In-situ Raman spectra were collected by using a 785 nm laser, which was focused through a 50 x objective. The laser power on the sample surface is ~ 20 mW. All spectra were calibrated by a standard silicon (111) wafer.

## Transmission Electron Microscope (TEM) and Electron Energy Loss Spectroscopy (EELS)

The TEM samples were prepared on an ThermoFisher Helios Nanolab 660 DualBeam focused ion beam (FIB) using the lift-out technique and transferred onto a TEM half-grid. The samples were cut perpendicular to the cleavage plane. TEM images were acquired on Thermo-Fisher TalosX at TEM mode with an accelerating voltage of 200 kV. In our STEM-EELS investigations, we utilized the ThermoFisher TalosX2 Transmission Electron Microscope (TEM) paired with a Gatan Continuum EELS system for high-resolution imaging. A cryo-holder (Model 205, Simple Origin cryo holder) was employed to mitigate electron beam-induced damage. The TalosX2 TEM was operated at an acceleration voltage of 200 kV. We adjusted the EELS system to select an energy dispersion of 0.05 eV/ch through the aperture. The alignment of the Zero-loss peak was executed, revealing a full width at half maximum (FWHM) of 1.1 eV. Subsequently, the Dual peak mode was activated to simultaneously monitor the O K-edge at 532 eV and the zero-loss peak. During the imaging phase, we adhered to the following parameters: a spot size of 6, a C2 aperture size of 50 μm, a camera length (CL) of 205 mm, and a convergence semi-angle of 7.5 mrad. The screen current and pixel dwell time were consistently set at 0.36 nA and 0.05 s respectively, with an energy dispersion of 50 meV/ch. To further minimize e-beam damage to the specimens, we adopted EELS mapping and integrated the signal over broader regions. Our analytical approach normalized EELS core-loss spectra against their corresponding zero-loss peak maximum values. A comparative EELS assessment between the leached and untouched samples focused on regions with a relative thickness near 0.25 mean free paths.

## Reporting summary

Further information on research design is available in the Nature Portfolio Reporting Summary linked to this article.

## Data availability

The data supporting this study's findings are available in the Article and its Supplementary Information. The datasets generated during and/or analyzed during the current study are available from the corresponding author upon reasonable request.

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

## Acknowledgements

H.Z. and F.S. thank the support from the Assistant Secretary for Energy Efficiency and Renewable Energy, Office of Vehicle Technologies of the US Department of Energy through the Advanced Battery Materials Research Program. J.L. and F.S. thank the support from the U.S. Department of Energy under contract DE-NE0009286. F.S. thanks the support from the National Science Foundation under Grant No. 2239690. Y.Y. and Y. H. thank the support from the Institutes of Energy and the Environment (IEE) Seed Grant Program at Pennsylvania State University. The co-authors acknowledge the use of the Penn State Materials Characterization Lab. The co-authors acknowledge Yongwen Sun and Zhiyu Zhang for their support on FIB sample preparation, cryo-STEM/EELS data collection, and analysis.

## Author contributions

H.Z. and F.S. conceived the idea and designed the experiments. H.Z. conducted materials synthesis and electrochemical tests. H.Z., Y.H., and J.L. contributed to the characterization. H.Z., Y.H., J.L., J.W., Z.L., Y.Y., and F.S. collaboratively analyzed the data; H.Z., J.W., and F.S., wrote the manuscript. All authors commented on the final manuscript.

## Competing interests

This work is submitted to US Patent Application on 07/05/2023 by The Penn State Research Foundation as Zhang, H.& Shi, F. U.S. patent application. PCT/US2023/022350. "Direct Electrochemical extraction of Lithium from ores." The other authors declare no competing interests.
