## [Peer Review File · Nature Communications]

Direct extraction of lithium from ores by electrochemical leachingREVIEWER COMMENTS

Reviewer #1 (Remarks to the Author):

The authors reported an electrochemical leaching process to directly extract lithium from α -phase spodumene using diluted acid (0.5M H₂SO₄) under room temperature. And design a 3D current collector to realize high feedstock loading and leaching current. Compared to traditional processes that utilize high temperatures and concentrated acid or alkali, this process proposed seems to be more environmentally and economically viable. Moreover, the promoter added, namely H₂O₂, is a simple and widely used chemical, presenting great potential for large scale application. However, the conclusions especially those related to the leaching mechanism with H₂O₂ are not well-supported. Considering the leaching performance and the scalability of the electrochemical leaching process reported, I recommend the acceptance of this work after major revision.

1. Oxygen evolution reaction is a competitive reaction in this process. According to Figure 1f and Figure 2b, it looks like the addition of H₂O₂ promotes not only the leaching of lithium but also the OER. In such circumstances, how can the authors ensure the leaching efficiency and Faradic Efficiency?

2. The leaching efficiency is highly related to the leaching potential applied. Can the authors clarify how they determine the leaching potential? Also, it is recommended to discuss the electrochemical leaching condition optimization section before discussing the leaching mechanism, which will make it easier to follow.

3. What is the exact role of H₂O₂ and the reaction pathway of H₂O₂? It was noticed that all LSV tests are conducted from 0.8 V to 1.1 V, yet the in-situ Raman test (Figure 2e) was performed from 0.2 V to 0.8 V. As the data show, the signal of O₂- surfaces when the leaching potential goes to 0.6 V and vanishes when the potential reaches 0.8 V, which, the authors claimed, was attributed to OER. This is really confusing and kinda contradictory. If so, the authors are expected to provide new LSV accordingly and make clear when the leaching reaction starts and what exactly happens when lifting the potentials.

4. As for the in-situ formation of H₂O₂, little data and discussion is provided. As discussed in the main text, 0.1 wt% H₂O₂ is expected to sustain the leaching efficiency yet how could the use of Au-GO-CF current collector facilitate this process and how can the amount of H₂O₂ be monitored? With Au-GO-CF current collector, when the leaching potentials increase from 0.8 V to 1.1 V, as presented in Figure 3e, what is the contribution of H₂O₂ formation? Also, from the figure, the GO-CF current collector enables similar output currents as Au-GO-CF.

5. Apart from characterizing the chemical composition of the leaching residual, it is better to monitor the gaseous products in situ to support the overall reaction equation presented in the Supporting Information (Page 38). In-situ DEMS could offer valuable information in this respect.

6. Legends are missing in Figure 3f, 3h.

Reviewer #2 (Remarks to the Author):

Review of NCOMMS-23-62173 "Direct extraction of lithium from ores by electrochemical leaching "

The manuscript describes a potentially interesting alternative to the recovery of lithium from solid state alpha phase spodumene. The work has the potential to bring forth an important advance in this research area, but before publication a few key areas might need to be

addressed:

- 1) An improvement of the current manuscript would be the inclusion of a more in-depth discussion about why the alpha-spodumene would lend itself to good leaching efficiencies, whereas the beta-phase undergoes a phase transformation, especially in the context of the beta phase being relatively more reactive.
- 2) I was particularly interested in learning more about the methodology by which the extraction is achieved. I found myself wondering what type of interaction causes this efficient extraction – The lithium extracted is presumably an ion but how is this done, at the chemical level, is it replaced by something else? This would be an important element to add to the discussion and to address throughout the manuscript – what are the chemical and physical transformations behind this observed leaching
- 3) Can this method be used to recover lithium from SC6, a common spodumene concentrate?
- 4) I realize that the authors have put in a great deal of effort in this area, but I wonder how reproducible this work is- error analysis on the yields.
- 5) Can the authors comment on similarities and differences between their methodology and electro dialysis recovery methods? Mater. Adv., 2021, 2, 1113-1138 has reviewed this area among others.
- 6) Given that the real ore-grade spodumene may include impurities and other elements, do impurities affect the quality of the leachate?
- 7) The O₂ dianion is stated on p9 as a metal stable species, but surely this is not always the case. Hydrogen peroxide is a relatively good oxidant for metals. What happens if hydrogen peroxide is introduced, and no bias is applied? Has this control been done, if yes, it needs to be prominently featured in the discussion
- 8) Hydrogen peroxide from commercial sources is often reagent grade and may contain a non-trivial amount of metal ions among other impurities- it is also reactive such that we ask our students to avoid metal needles in its handling. Have the authors excluded the possibility of metal ion introduction through the addition of H₂O₂?
- 9) Lastly, I was unable to discern what specific particle/crystal size was being utilized in these studies? Is this size uniform? How is this controlled and how does it affect the leaching process efficiency?

Thank you for the opportunity to review this work and I look forward to a revision.

The authors are grateful to the reviewers for their constructive comments on the original manuscript. All reviewers' comments were carefully considered and implemented/responded in the revised manuscript. A detailed response is given for each comment. We italicized the reviewers' comments and suggestions and highlighted our responses in purple for clarity.

Reviewer #1 (Remarks to the Author):

The authors reported an electrochemical leaching process to directly extract lithium from α -phase spodumene using diluted acid (0.5M H₂SO₄) under room temperature. And design a 3D current collector to realize high feedstock loading and leaching current. Compared to traditional processes that utilize high temperatures and concentrated acid or alkali, this process proposed seems to be more environmentally and economically viable. Moreover, the promoter added, namely H₂O₂, is a simple and widely used chemical, presenting great potential for large scale application. However, the conclusions especially those related to the leaching mechanism with H₂O₂ are not well-supported. Considering the leaching performance and the scalability of the electrochemical leaching process reported, I recommend the acceptance of this work after major revision.

Thank you for your encouraging and constructive comments. We appreciate your encouraging feedback and suggestions for improving the quality and clarity of the manuscript. The point-by-point response to the referee's comments is given below.

1. Oxygen evolution reaction is a competitive reaction in this process. According to Figure 1f and Figure 2b, it looks like the addition of H₂O₂ promotes not only the leaching of lithium but also the OER. In such circumstances, how can the authors ensure the leaching efficiency and Faradic Efficiency?

Our response:

Yes, the decomposition of additive H₂O₂ will contribute to the Faraday current of the OER reaction.

As shown in Figure 1f and Figure 2b (we normalized the range of the x-axis, so it is more obvious), there is no electrochemical reaction other than OER, without spodumene. After adding H₂O₂, the electrochemical oxidation peak shifts from 1.1-1.2 V to 0.95 V vs. SCE. In Figure 2c,

we find that the critical concentration of H₂O₂ is ~0.1 wt.%, below this concentration, there's no leaching reaction activated. In the meanwhile, a higher concentration of H₂O₂ (e.g. 1wt%) will consume more electrons for the OER reaction. Hence, we choose a moderate concentration of 0.5 wt.% in the high throughput test.

Figure R1-1. **1f**, electrochemical leaching test using linear sweep voltammetry (LSV) on α and β -phase spodumene and a graphite electrode with conductive carbon and Nafion binder in 0.5 M H₂SO₄ aqueous electrolyte at a scan rate of 0.5 mV s⁻¹. **2b**, Electrochemical leaching test with linear sweep voltammetry (LSV) on α and β -phase spodumene in 0.5 M H₂SO₄ aqueous electrolyte with 0.5 wt.% H₂O₂ as a promoter at a scan rate of 0.5 mV s⁻¹. **2c**, Electrochemical leaching performance with LSV using various H₂O₂ concentrations in 0.5 M H₂SO₄ aqueous electrolyte at a scan rate of 0.5 mV s⁻¹.

As for leaching efficiency and Faradaic efficiency:

(1) The **leaching efficiency** is calculated as the ratio of the lithium ions in leachant (the electrolyte in this work) to the total Li content from pristine α -phase spodumene.

Both values are extracted from the ICP-AES experiment, which will not be influenced by the electrochemical signal.

$$\text{Leaching efficiency} = \frac{\text{Content of Li in leachant}}{\text{Total content of Li from spodumene}}$$

(2) **Faradaic efficiency** (FE) is calculated by the charge for electrochemical leaching over the total charge passed. The total charge passed comprises two parts: the charge for electrochemical leaching (Q_{Leaching}) and the charge consumed by side reactions (Q_{OER}). As shown in the schematic below (updated Fig. S22a).

Supplementary Fig. 22. The calculation of Faradaic efficiency. a, the schematic shows the charge for leaching (blue) and side reactions (red). The charge Q_{Leaching} and Q_{OER} are calculated by the integration of the current i_{Leaching} (leaching current with the spodumene) and i_{OER} (the background current without any active materials).

For example, in Fig. 3h, we integrated the charge passed without the active materials as the charge consumed by the side reactions (Q_{OER} , the charge passed without any spodumene particles, indicating the charge consumed by side reactions). We also calculated the charge passed with active materials as the total charge passed ($Q_{\text{leaching}} + Q_{\text{OER}}$, the charge passed with active materials α -spodumene), the charge for electrochemical leaching is Q_{leaching} . The FE at 0.95 V vs. SCE indicates that 71.5% of the electrons are utilized to extract lithium from the spodumene, and the side reactions consume 28.5 % of the electrons.

$$\text{The Faradaic efficiency is calculated by}^1: FE = \frac{Q_{\text{Leaching}}}{Q_{\text{Leaching}} + Q_{\text{OER}}}$$

Change to manuscript:

To clarify this, we modified the manuscript's main context, **Figure. S22**, and **figure caption**, so the audience can better understand the measure of FE and leaching efficiency.

Faradaic efficiency (FE) is usually expressed as a percentage, and it is defined as the amount of collected product to the theoretical amount of product that can be produced from the total charge passed². Figure S22 shows the schematic of calculation and FE under different voltages. (Page 13, line 22, main context)

Leaching efficiency is calculated as the ratio of the lithium ions in leachant (the electrolyte in this work) to the total Li content from pristine α -phase spodumene. It's derived based on ICP result, which is not influenced by the electrochemical signal (Page 14, line 8, main context)

Fig. S22 and its captions are updated.

Supplementary Fig. 22. The calculation of Faradaic efficiency. **a**, the schematic shows the charge for leaching (blue) and side reactions (red). The charge Q_{Leaching} and Q_{OER} are calculated by the integration of the current i_{Leaching} (leaching current with the spodumene) and i_{OER} (the background current without any active materials). **b**, the charge passed at various leaching potentials by holding for 30 mins (without spodumene). **c**, the charge passed at various leaching potentials by holding for 30 mins (with 1 g spodumene). The hollow circles denote the total charge passed without active materials (Q_{OER} , indicating the charge consumed by side reactions), and the filled circles denote the total charge passed with active materials ($Q_{\text{Leaching}} + Q_{\text{OER}}$).

2. The leaching efficiency is highly related to the leaching potential. Can the authors clarify how they determine the leaching potential? Also, it is recommended to discuss the electrochemical leaching condition optimization section before discussing the leaching mechanism, which will make it easier to follow.

Our response:

Yes, both leaching efficiency and Faradaic efficiency are related to the applied potential. The leaching potential reaches the maximum of $\sim 90\%$ when the applied potential is > 0.95 V vs. SCE after 12hrs. Faradaic efficiency has a “volcano shape” relationship with the applied potential, which reaches the maximum of 71.5% at 0.95 V vs. SCE. Lower applied potentials are insufficient to trigger lithium leaching reactions, while higher potentials lead to more side reactions. We also monitored the H_2O_2 concentration **with the microelectrode** method in Figure 3h. We observe a quick drop of H_2O_2 concentration above 1V. Hence, we determine the leaching potential based on the highest Faradaic efficiency (0.95 V) and local H_2O_2 concentration. We added some discussion to clarify this point.

Updated Fig. S21d Localized H_2O_2 concentration as a function of leaching voltage.

We thank the reviewer for the constructive comment about the structure. The logic of this manuscript is (1) demonstrating that lithium can be electrochemically leached from the spodumene; (2) adding a leaching promoter to lower the energy barrier; (3) understanding the leaching mechanism and the role of the leaching promoter; (4) design a multi-functional current collector for scale-up leaching and leaching condition optimization.

Change to manuscript:

We added discussion into the main context, including how we selected the optimized leaching potential by balancing the Faradaic efficiency, leaching efficiency, and minimizing the side reactions. We also updated **Figure. 3** and **Fig. S21** for clarity.

The highest Faradaic efficiency occurred between 0.9 V – 1.0 V vs. SCE, as lower potentials (region I, <0.9 V vs. SCE) are not sufficient to trigger lithium leaching reactions, while higher potentials (region III, > 1 V) lead to more side reactions (oxygen evolution, decomposition of H_2O_2), and localized H_2O_2 concentration suddenly dropped after 1 V, as shown in Fig. S22d. In contrast, the highest leaching efficiency of GO-CF and CF is 64.9 % and 43.2 % (at 1 V vs. SCE), respectively. (Page 13, line 30, main context)

It can be found that 0.95 V vs. SCE is the optimized potential for achieving the highest leaching efficiency, where we reached 92.2% after 12-hour electrochemical leaching. At the lower leaching potential (0.9 V vs. SCE), only 24.3% leaching efficiency is achieved after 20 hours. The leaching rate is constant, showing that the reaction proceeds slowly. At the higher potential region (1.0 V vs. SCE), the leaching speed is fast at the beginning (from 0-8 hours) but gradually slows down and doesn't reach the maximum leaching efficiency until 20 hours, resulting from more side reactions such as the OER and the decomposition of the H_2O_2 . (Page 14, line 17, main context)

0.95 V vs. SCE is selected as the optimized leaching potential carefully balancing the conditions that promote high Faradaic efficiency and leaching efficiency with minimal side reactions. At this leaching potential, the Faradaic efficiency reaches its peak of 71.5% due to the efficient utilization of electrons for the leaching reaction rather than side reactions. Additionally, the leaching efficiency also achieves the highest of 92.2%, as higher potentials led to faster initial reactions but more side reactions, ultimately lowering the leaching efficiency. This leaching potential can make full use of the energy while achieving the highest leaching degree. (Page 14, line 26, main context)

Fig. S21 is updated with the localized H_2O_2 concentration corresponding to Fig. 3f and 3h. The change in the localized H_2O_2 concentration can be easily visualized. The side reactions became drastic after 1 V vs. SCE.

Supplementary Fig. 21. Cell configuration for the scale-up electrochemical test and in-situ monitoring H_2O_2 with microelectrode. **a**, side-view **b**, top-view. The working electrode (WE) is GO-modified porous carbon felt, the counter electrode (CE) is graphite rod or carbon paper, and the reference electrode (RE) is saturated calomel electrode (SCE). A microelectrode is used as the second working electrode for in-situ monitoring H_2O_2 concentration. **c**, localized H_2O_2 concentration as a function of leaching time by holding at 0.95 V vs. SCE, corresponding to the concentration in Fig. 3f. **d**, localized H_2O_2 concentration as a function of leaching voltage, corresponding to the concentration in Fig. 3h.

Figure 3 | High throughput current collector design and leaching conditions optimization. a, The morphology of the GO-modified current collector. **b**, Electrochemical leaching performance with various binders (Nafion, PVDF, CMC/SBR). **c**, Distribution of Au nanoparticles. **d**, high-resolution transmission electron microscope (HRTEM) image of a single Au particle. **e**, Current comparison of different current collectors hold from 0.8 V – 1.1 V vs. SCE. **f**, Stability test of different current collectors by holding at 0.95 V vs. SCE. **g**, The in-situ monitor of H_2O_2

concentration by a 34 μm microelectrode, inset: the linear fitting of the current density as a function of H_2O_2 concentration. **h**, Faradaic efficiency of different current collectors at various leaching potentials. **i**, Leaching efficiency as a function of leaching potentials using Au-GO-CF current collector. For the scale-up electrochemical leaching, 1 g spodumene is suspended in the electrolyte. Scale bars: a, 100 μm . c, 500 nm. d, 2nm.

3. *What is the exact role of H_2O_2 and the reaction pathway of H_2O_2 ? It was noticed that all LSV tests are conducted from 0.8 V to 1.1 V, yet the in-situ Raman test (Figure 2e) was performed from 0.2 V to 0.8 V. As the data show, the signal of O_2 surfaces when the leaching potential goes to 0.6 V and vanishes when the potential reaches 0.8 V, which, the authors claimed, was attributed to OER. This is really confusing and kinda contradictory. If so, the authors are expected to provide new LSV accordingly and make clear when the leaching reaction starts and what exactly happens when lifting the potentials.*

Our response:

Thank you for pointing this confusion out. As stated in the original manuscript:

“A home-built three-electrode cell uses a Ag wire counter electrode and a Ag wire reference electrode”. (Page 9, line 26)

The voltage in the original Fig. 2g is actually measured vs. Ag/Ag⁺. In the rest of this work's electrochemistry test, we are using a saturated calomel electrode (SCE) for the measurement. The difference between SCE and Ag wire is around 0.2V, as shown in the OCV measurement below.

The OCV test between Ag wire vs. SCE reference electrode.

After normalizing the reference electrode, the Raman signal of Li_2O_2 starts to show above~0.8V (updated below).

We updated both text and figures with normalized potential to vs. SCE as below:

Figure R1-3. **a**, the new In-situ Raman spectra. The voltage is normalized to vs. SCE, consistent with the rest of the voltage mentioned in the manuscript. **b**, LSV scan of the α -spodumene in 0.5 M H_2SO_4 with 0.5 wt.% H_2O_2 .

Change to the manuscript:

We modified the discussion in the main context, **Fig. 2**, and **its caption**, so the readers can understand our reference electrode and the reaction potentials.

“As shown in Fig. 2g, during the electrochemical leaching, as the leaching voltage increases to 0.8 V vs. SCE, there starts to show the signal of Li_2O_2 (258 cm^{-1}), demonstrating that peroxide is an intermediate product during the electrochemical leaching. The signal starts to weaken when voltage is above 1 V vs. SCE, mostly due to the oxygen evolution reactions.” (page 9, line 30, main context)

Figure 2 and its caption are modified to vs. SCE.

Figure 2 | Thermodynamic and kinetics study for electrochemical leaching with H_2O_2 promoter **a**, Gibbs free energy change prediction of spodumene leaching reactions with/without H_2O_2 calculated by HSC chemistry software (Version 10.1). **b**, Electrochemical leaching test with linear sweep voltammetry (LSV) on α and β -phase spodumene in 0.5 M H_2SO_4 aqueous electrolyte with 0.5 wt.% H_2O_2 as a promoter at a scan rate of 0.5 mV s^{-1} . **c**, Electrochemical leaching performance with LSV using various H_2O_2 concentrations in 0.5 M H_2SO_4 aqueous electrolyte at a scan rate of 0.5 mV s^{-1} . $\sim 20\text{mg}$ spodumene particles are coated on a graphite rod with conductive carbon and Nafion binder **d**, X-ray diffraction (XRD) patterns of pristine and leached spodumene as a function of electrochemical leaching time. **e**, Lattice structure of α -phase spodumene imaged

by high-resolution transmission electron microscope (HRTEM). Insets: corresponding FFT pattern. **f**, The X-ray photoelectron spectroscopy (XPS) spectra centered on Al 2p before and after leaching. **g**, 3-electrode in-situ Raman spectra as a function of leaching time. The counter electrode is silver wire, and the voltage is converted to voltage vs. SCE. Scale bar: 10 nm.

4. As for the in-situ formation of H₂O₂, little data and discussion is provided. As discussed in the main text, 0.1 wt% H₂O₂ is expected to sustain the leaching efficiency yet how could the use of Au-GO-CF current collector facilitate this process and how can the amount of H₂O₂ be monitored? With Au-GO-CF current collector, when the leaching potentials increase from 0.8 V to 1.1 V, as presented in Figure 3e, what is the contribution of H₂O₂ formation? Also, from the figure, the GO-CF current collector enables similar output currents as Au-GO-CF.

Our response:

Thank you for pointing out this potential confusion. We monitored the local concentration of H₂O₂ with **microelectrode**, as shown below (re-scaled).

Updated Fig S21c, localized H₂O₂ concentration as a function of leaching time by holding at 0.95 V vs. SCE, corresponding to the concentration in Fig. 3f. **Updated Fig. S21d**, localized H₂O₂ concentration as a function of leaching voltage, corresponding to the concentration in Fig. 3h.

The data above shows that the best H₂O₂ concentration retention is with a catalyst-modified current collector. This high concentration retention is essential to maintaining the high leaching efficiency when the original added H₂O₂ is consumed during the long-term test. Localized formation of H₂O₂ on Au-GO-CF current collector (cathode side) will minimize the cross-over side reaction on the anode, during the leaching process.

The monitoring of localized H₂O₂ concentration is mentioned in the original main context and Supplementary Note 2, quoted below:

*“In this study, we use a glassy carbon **microelectrode** (34 μm diameter) to measure the H₂O₂ concentrations (details in Supplementary Note 2 and Fig. S21). By holding the potential below the H₂O₂ reduction potential (-0.65 V vs. SCE), H₂O₂ is continuously reduced, and the current density is proportional to the H₂O₂ concentrations. The inset of Fig. 3g presents the linear fitting of the current density as a function of H₂O₂ concentration, and the coefficient of determination (R^2) is > 0.99, showing a good correlation between the H₂O₂ concentration and reduction current density. The H₂O₂ concentration in Fig. 3f is determined by the **microelectrode**, and it can be seen the leaching current is directly related to the H₂O₂ concentration.”* (Page 13, line 9, main context)

*“There are several methods for measuring H₂O₂ concentration, such as titration, refractory meter, or commercial H₂O₂ strips. However, these methods only provide the static bulk solution's concentration. By holding the potential below the H₂O₂ reduction potential (-0.65 V vs. SCE), the localized H₂O₂ is reduced by the microelectrode, and the current density is proportional to the H₂O₂ concentrations (Fig. 3g), as reported previously. It should be noted that for macroelectrodes, cyclic voltammetry scans result in a peak current followed by a diffusional tail, showing the reactant must be transported to the electrode by diffusion. In contrast, CV for **microelectrodes** results in a current plateau (like a rotating ring-disk electrode) due to a steady-state response from the 3-D diffusion.”* (Supplementary Note 2)

Au nanoparticles are widely reported as good catalysts for the electrochemical synthesis of hydrogen peroxide³⁻⁵. In the original main context, we have discussed Au-GO-CF current collector has the best current retention, even though the initial current is similar to GO-CF. As below:

“The H₂O₂ concentration in Fig. S21c is determined by the microelectrode, and it can be seen the leaching current is directly related to the H₂O₂ concentration. As the H₂O₂ concentration decreases in Fig. S21c, the leaching current in Fig. 3f also decreases. The Au-GO-CF current collector has the best current retention because it can retain the highest H₂O₂ concentration of 0.30 wt.% after 20 hours of leaching.” (Page 13, line 15, main context)

“The highest Faradaic efficiency occurred between 0.9 V – 1.0 V vs. SCE, as lower potentials (region I, <0.9 V vs. SCE) are not sufficient to trigger lithium leaching reactions, while higher potentials (region III, > 1 V) lead to more side reactions (oxygen evolution, decomposition of

H₂O₂), and localized *H₂O₂* concentration suddenly dropped after 1 V, as shown in Fig. S22d.”
(Page 13, line 28, main context)

We are not quite clear about the “*contribution of H₂O₂ formation*”. If you are asking about the contribution to the total leaching current, it is hard to separate the contribution from the oxidation of H₂O, the oxidation of H₂O₂, the electrochemical leaching of spodumene, and the formation of H₂O₂. It is reasonable that GO-CF and Au-GO-CF have very similar initial leaching currents since their surface area is very close. However, Au-GO-CF has a better leaching current retention owing to the contribution of Au catalysts, as shown below (Fig. 3f).

Fig 3f, Stability test of different current collectors by holding at 0.95 V vs. SCE.

Change to manuscript:

Fig. S21 and Fig. 3 are updated to visualize the evolution of H₂O₂ concentrations.

Fig. S21 is updated with the localized H₂O₂ concentration corresponding to Fig. 3f and 3h. The change in the localized H₂O₂ concentration can be easily visualized. The side reactions became drastic after 1 V vs. SCE.

Supplementary Fig. 21. Cell configuration for the scale-up electrochemical test and **in-situ** monitoring H₂O₂ with **microelectrode**. **a**, side-view **b**, top-view. The working electrode (WE) is GO-modified porous carbon felt, the counter electrode (CE) is graphite rod or carbon paper, and the reference electrode (RE) is saturated calomel electrode (SCE). A microelectrode is used as the second working electrode for in-situ monitoring H₂O₂ concentration. **c**, localized H₂O₂ concentration as a function of leaching time by holding at 0.95 V vs. SCE, corresponding to the concentration in Fig. 3f. **d**, localized H₂O₂ concentration as a function of leaching voltage, corresponding to the concentration in Fig. 3h.

Figure 3 | High throughput current collector design and leaching conditions optimization. **a**, The morphology of the GO-modified current collector. **b**, Electrochemical leaching performance with various binders (Nafion, PVDF, CMC/SBR). **c**, Distribution of Au nanoparticles. **d**, high-resolution transmission electron microscope (HRTEM) image of a single Au particle. **e**, Current comparison of different current collectors hold from 0.8 V – 1.1 V vs. SCE. **f**, Stability test of different current collectors by holding at 0.95 V vs. SCE. **g**, The in-situ monitor of H_2O_2 concentration by a $34\ \mu\text{m}$ microelectrode, inset: the linear fitting of the current density as a function of H_2O_2 concentration. **h**, Faradaic efficiency of different current collectors at various leaching potentials. **i**, Leaching efficiency as a function of leaching potentials using Au-GO-CF current collector. For the scale-up electrochemical leaching, 1 g spodumene is suspended in the electrolyte. Scale bars: a, $100\ \mu\text{m}$. c, $500\ \text{nm}$. d, $2\ \text{nm}$.

Reference

3 Landon, P. et al. Direct synthesis of hydrogen peroxide from H₂ and O₂ using Pd and Au catalysts. *Physical Chemistry Chemical Physics* 5, 1917-1923 (2003). <https://doi.org/10.1039/B211338B>

4 Landon, P., Collier, P. J., Papworth, A. J., Kiely, C. J. & Hutchings, G. J. Direct formation of hydrogen peroxide from H₂/O₂ using a gold catalyst. *Chemical Communications*, 2058-2059 (2002). <https://doi.org/10.1039/B205248M>

5 Pizzutilo, E. et al. Gold–Palladium Bimetallic Catalyst Stability: Consequences for Hydrogen Peroxide Selectivity. *ACS Catalysis* 7, 5699-5705 (2017). <https://doi.org/10.1021/acscatal.7b01447>

5. Apart from characterizing the chemical composition of the leaching residual, it is better to monitor the gaseous products in situ to support the overall reaction equation presented in the Supporting Information (Page 38). In-situ DEMS could offer valuable information in this respect.

Our response:

Thanks for the constructive comments. Differential Electrochemical Mass Spectrometry (DEMS) is a powerful analytical technique that combines electrochemistry and mass spectrometry to study the mechanisms of electrochemical reactions and the species involved. In previous studies, researchers identified the origin of gaseous phase products by isotope labeling^{6,7}. However, in our case, there are several practical issues:

1. The origin of O₂ is too complicated. In our leaching system, we have H₂O, H₂O₂, spodumene (LiAlSi₂O₆), and the injected O₂ (to form H₂O₂). In this case, it's very difficult to trace the origin of O₂ products.

2. The cost of isotope labeling. If we want to trace the origin of O₂ products, we need to label the O in the reactants. For H₂¹⁸O₂ (90 atom %), the price is \$3,030.00 per gram, and for H₂¹⁸O (99 atom %), it is \$1,700 per gram (99 atom %). Considering the volume of the reactant, the overall cost is too much. (Price data is retrieved from Sigma-Aldrich)

3. The leaching is done in an open system. There is the interference of O₂ from the surrounding atmosphere.

We believe that combining the local concentration of additive H₂O₂, Li⁺ concentration in the electrolyte, and current contribution from OER and electro-leaching would provide sufficient information to understand the mechanism of electrochemical leaching.

References:

6 Xu, G. et al. The Formation/Decomposition Equilibrium of LiH and its Contribution on Anode Failure in Practical Lithium Metal Batteries. *Angew Chem Int Ed Engl* 60, 7770-7776 (2021). <https://doi.org:10.1002/anie.202013812>

7 Fang, C. et al. Quantifying inactive lithium in lithium metal batteries. *Nature* 572, 511-515 (2019). <https://doi.org:10.1038/s41586-019-1481-z>

6. Legends are missing in Figure 3f, 3h.

Our response:

Thanks for the suggestions. We updated the legends of **Figure 3** for clarity.

Change to the manuscript:

Figure 3f and 3h are updated with the legend. We also moved the right y-axis of 3f and 3h to Fig S21 so the audience can clearly visualize the change in the H₂O₂ concentration. The caption is also updated.

Figure 3f and 3h are updated.

Figure 3 | High throughput current collector design and leaching conditions optimization. **a**, The morphology of the GO-modified current collector. **b**, Electrochemical leaching performance with various binders (Nafion, PVDF, CMC/SBR). **c**, Distribution of Au nanoparticles. **d**, high-resolution transmission electron microscope (HRTEM) image of a single Au particle. **e**, Current comparison of different current collectors hold from 0.8 V – 1.1 V vs. SCE. **f**, Stability test of different current collectors by holding at 0.95 V vs. SCE. **g**, The in-situ monitor of H_2O_2 concentration by a $34\ \mu\text{m}$ microelectrode, inset: the linear fitting of the current density as a function of H_2O_2 concentration. **h**, Faradaic efficiency of different current collectors at various leaching potentials. **i**, Leaching efficiency as a function of leaching potentials using Au-GO-CF current collector. For the scale-up electrochemical leaching, 1 g spodumene is suspended in the electrolyte. Scale bars: **a**, $100\ \mu\text{m}$. **c**, $500\ \text{nm}$. **d**, $2\ \text{nm}$.

Fig. S21 and its caption are also modified.

Supplementary Fig. 21. Cell configuration for the scale-up electrochemical test and **in-situ** monitoring H₂O₂ with **microelectrode**. **a**, side-view **b**, top-view. The working electrode (WE) is GO-modified porous carbon felt, the counter electrode (CE) is graphite rod or carbon paper, and the reference electrode (RE) is saturated calomel electrode (SCE). A microelectrode is used as the second working electrode for in-situ monitoring H₂O₂ concentration. **c**, localized H₂O₂ concentration as a function of leaching time by holding at 0.95 V vs. SCE, corresponding to the concentration in Fig. 3f. **d**, localized H₂O₂ concentration as a function of leaching voltage, corresponding to the concentration in Fig. 3h.

Reviewer #2 (Remarks to the Author):

The manuscript describes a potentially interesting alternative to the recovery of lithium from solid state alpha phase spodumene. The work has the potential to bring forth an important advance in this research area, but before publication a few key areas might need to be addressed:

Thank you for your encouraging and constructive comments. We appreciate your encouraging feedback and suggestions for improving the manuscript. The point-by-point response to the referee's comments is given below.

1) An improvement of the current manuscript would be the inclusion of a more in-depth discussion about why the alpha-spodumene would lend itself to good leaching efficiencies, whereas the beta-phase undergoes a phase transformation, especially in the context of the beta phase being relatively more reactive.

Our response:

Thanks for the reviewer's comments. β -phase is more reactive and it's easier to electrochemically leach lithium out of β -phase than α -phase. As shown in Fig. R2-2, no matter whether with/without the leaching promoter, the leaching current of the β -phase is always higher than that of the α -phase. This indicates that it's still easier to extract lithium from the β -phase than the α -phase. However, the phase transition from α -phase to β -phase is energy-intensive (1100 °C calcination for 10 hours). Hence, we put our emphasis on α -phase so this method is more eco-friendly than the traditional leaching method.

Figure R2-1. **1f**, electrochemical leaching test using linear sweep voltammetry (LSV) on α and β -phase spodumene and a graphite electrode with conductive carbon and Nafion binder in 0.5 M H_2SO_4 aqueous electrolyte at a scan rate of 0.5 mV s^{-1} . **2b**, Electrochemical leaching test with

linear sweep voltammetry (LSV) on α and β -phase spodumene in 0.5 M H₂SO₄ aqueous electrolyte with 0.5 wt.% H₂O₂ as a promoter at a scan rate of 0.5 mV s⁻¹.

Change to the manuscript:

We added a sentence to the original main context, to emphasize the significance of direct leaching using α -phase spodumene.

Given that α -phase spodumene represents its natural state and the phase transition to β -phase is energy-intensive, our research focuses on the α -phase to ensure that the leaching process is environmentally sustainable. (Page 11, line 18, main context)

2) I was particularly interested in learning more about the methodology by which the extraction is achieved. I found myself wondering what type of interaction causes this efficient extraction – The lithium extracted is presumably an ion but how is this done, at the chemical level, is it replaced by something else? This would be an important element to add to the discussion and to address throughout the manuscript – what are the chemical and physical transformations behind this observed leaching

Our response:

As the lithium ions are electrochemically leached out, the O₂²⁻ inside the lattice is oxidized to a higher valence to maintain electrical neutrality. This phenomenon is very similar to the cathode materials in lithium-ion batteries (LIBs). The electrochemical extraction of Li in metal oxide framework is well-studied in lithium-ion battery cathode materials, such as LiNi_xCo_yMn_zO₂ (NCM)⁸, LiM₂O₄ (LMO)⁹ and Li_{0.2}(LiNi_xCo_yMn_z)O₂ (Li-rich) cathode¹⁰⁻¹². During the charging stage (de-lithiation), LMO framework loses electrons, and lithium ions (positive charge) leave their origin sites, As shown in Fig. R2-2a, for LMO and Li-rich cathode, there are two stages to maintaining the electrical neutrality: (i) the oxidation of the transition metal element; (ii) the oxidation of O from O²⁻ → O₂²⁻ → O₂⁻ → O₂. Fig. R2-2b shows the TEM image of the charged state Li-rich cathode. As lithium leaves its original sites with oxygen evolution, the transition metal framework shrinks.

However, in spodumene ores (LiAlSi₂O₆), the Si and Al are stable elements with barely changed valence states. Higher O valence in leached spodumene means the O gets oxidation to keep the system electrically neutral. ICP-AES results (Supplementary Table 2) show over 90% of

lithium has been successfully leached out for both phases. As shown in Fig. S12 SEM-EDS analysis on leached residue shows the leached spodumene still possesses the Al:Si ratio of 1:2, showing the silicate framework is intact, consistent with the XRD results.

Figure R2-2. **a**, charge/discharge in Li-Rich cathodes¹⁰, **b**, STEM images before and after the charging plateau along the [010] direction showing loss of clearly defined TM–TM dumbbells arising from honeycomb ordering within the TM layer¹¹.

Supplementary Table 2. The lithium content of α -phase and β -phase spodumene by ICP-AES.

Sample name	Li ₂ O (wt. %)	Al ₂ O ₃ (wt. %)	SiO ₂ (wt. %)
α -phase pristine	7.03	26.3	65.0
α -phase leached	0.75	28.1	70.6
β -phase pristine	7.20	26.6	66.0
β -phase leached	0.02	26.0	66.8

Supplementary Fig. S12. Post-leaching SEM characterization. SEM images of leached **a**, α -phase spodumene and **b**, β -phase spodumene. EDS spectrum and the elemental ratio of **c**, α -phase spodumene, and **d**, β -phase spodumene. Scale bar: 10 μm .

To further demonstrate O_2^{2-} is an intermediate product, we used in-situ Raman, as shown in Fig. R2-3, during the electrochemical leaching, as the leaching potential increases to 0.8 V vs. SCE, there starts to show the signal of Li_2O_2 (258 cm^{-1}), demonstrating that peroxide is an intermediate product during the electrochemical leaching. For holding at a constant potential of 0.8 V vs. SCE, when leaching time increases to 6 mins, Li_2O_2 can be observed. It is a meta-stable species, and it is hard to observe via ex-situ characterization tools. Hence, the addition of H_2O_2 increases the concentration of the intermediate product and facilitates the whole reaction.

Supplementary Fig. R2-3. In-situ Raman spectra as a function of voltage (top) and the evolution of the spectra versus leaching time while holding at 0.8 V (bottom). Voltage is measured vs. SCE.

Reference

8 Wandt, J., Freiberg, A. T. S., Ogrodnik, A. & Gasteiger, H. A. Singlet oxygen evolution from layered transition metal oxide cathode materials and its implications for lithium-ion batteries. *Materials Today* 21, 825-833 (2018). <https://doi.org/10.1016/j.mattod.2018.03.037>

9 Nomura, Y. et al. Dynamic imaging of lithium in solid-state batteries by operando electron energy-loss spectroscopy with sparse coding. *Nat Commun* 11, 2824 (2020). <https://doi.org/10.1038/s41467-020-16622-w>

10 Grenier, A. et al. Nanostructure Transformation as a Signature of Oxygen Redox in Li-Rich 3d and 4d Cathodes. *J Am Chem Soc* 143, 5763-5770 (2021). <https://doi.org/10.1021/jacs.1c00497>

11 House, R. A. et al. First-cycle voltage hysteresis in Li-rich 3d cathodes associated with molecular O₂ trapped in the bulk. *Nature Energy* 5, 777-785 (2020). <https://doi.org/10.1038/s41560-020-00697-2>

12 Luo, K. et al. Charge-compensation in 3d-transition-metal-oxide intercalation cathodes through the generation of localized electron holes on oxygen. *Nat Chem* 8, 684-691 (2016). <https://doi.org/10.1038/nchem.2471>

3) *Can this method be used to recover lithium from SC6, a common spodumene concentrate?*

Our response:

This is a very interesting topic related to practical application. We are working with some industrial partners to apply this technology to industrial-grade raw materials.

We believe this method can work for SC6, as the main component is also spodumene.

4) *I realize that the authors have put in a great deal of effort in this area, but I wonder how reproducible this work is- error analysis on the yields.*

Our response:

Thank you for your insightful feedback. The experiments were conducted using spodumene inputs at the gram scale (1 g), which inherently minimizes potential errors compared to the experiments using milli-gram or even micro-gram reactants. To further ensure the reliability of our findings, we replicated each experimental condition three times. The ICP-AES results were then used to calculate both the Faradaic efficiency and the leaching efficiency, to ensure the error analysis. The deviation of leaching efficiency is < 2 % and the deviation of the FE is < 3%. We updated Figure 3 and Table. S5 to clarify this point.

Change to manuscript:

We repeated the electrochemical leaching with the same conditions under 0.9 V – 1 V. The results show good reproducibility. We updated Fig. 3h with the error bar and updated with Table. S5 with the measured ICP-AES results.

Figure 3h is updated with repeated Faradaic efficiency at 0.9, 0.95, and 1 V vs. SCE. The error bar shows the measurement variation is < 3%.

Table S5 is updated with repeated conditions.

Supplementary Table 5. Chemical composition of the leachant by ICP-AES and the leaching efficiencies.

Leaching potential (V vs. SCE)	Feedstock (g)	Theoretical concentration (ppm)	Concentration by ICP (ppm)	Leaching efficiency (%)
No voltage (24 h)	1.0	671	0.35	0.05
0.9	1.0	671	322	48.08
0.9	1.0	671	310	46.20
0.9	1.0	671	314	46.79

0.95	1.0	671	618	92.12
0.95	1.0	671	614	91.51
0.95	1.0	671	607	90.46
1.0	1.0	671	604	91.22
1.0	1.0	671	609	90.76
1.0	1.0	671	610	90.90

5) Can the authors comment on similarities and differences between their methodology and electro dialysis recovery methods? Mater. Adv., 2021, 2, 1113-1138 has reviewed this area among others.

Our response:

In Mater. Adv., 2021, 2, 1113-1138¹³, it cited two papers regarding the electro dialysis recovery of lithium.

The similarities are that all the works used the electric field as the driving force for elemental recovery. Compared to other methods using chemical concentration or heating, electrochemical methods are more efficient and environmentally friendly.

In our study, we used an electrochemical method to extract lithium ions out of the solid-state ores into the aqueous phases. And in their studies, the authors first chemically dissolved the lithium ions from solid-state lithium waste, and then used the electrochemical method as a way of separation, purification, and enrichment^{14,15}.

Reference

13 Petersen, H. A., Myren, T. H. T., O’Sullivan, S. J. & Luca, O. R. Electrochemical methods for materials recycling. Materials Advances 2, 1113-1138 (2021). <https://doi.org:10.1039/d0ma00689k>

14 Song, Y. & Zhao, Z. Recovery of lithium from spent lithium-ion batteries using precipitation and electro dialysis techniques. Separation and Purification Technology 206, 335-342 (2018). <https://doi.org:10.1016/j.seppur.2018.06.022>

15 Li, Z., He, L., Zhao, Z. w., Wang, D. & Xu, W. Recovery of Lithium and Manganese from Scrap LiMn₂O₄ by Slurry Electrolysis. ACS Sustainable Chemistry & Engineering 7, 16738-16746 (2019). <https://doi.org:10.1021/acssuschemeng.9b04127>

6) *Given that the real ore-grade spodumene may include impurities and other elements, do impurities affect the quality of the leachate?*

Our response:

For spodumene, the majority of the gangue mineral/ impurity is quartz (SiO₂)¹⁶, which is chemically stable.

Ore-grade spodumene may include some other metal ions, and they will be easily dissolved as they have a higher redox potential than lithium. This will impact the quality of the leachant and more efforts need to be made for purification. We're going to collaborate with our industrial partner and study more effects of the ore-grade raw materials.

Reference:

16 Kundu, T., Rath, S. S., Das, S. K., Parhi, P. K. & Angadi, S. I. Recovery of lithium from spodumene-bearing pegmatites: A comprehensive review on geological reserves, beneficiation, and extraction. Powder Technology 415 (2023). <https://doi.org:10.1016/j.powtec.2022.118142>

7) *The O₂ dianion is stated on p9 as a metal stable species, but surely this is not always the case. Hydrogen peroxide is a relatively good oxidant for metals. What happens if hydrogen peroxide is introduced, and no bias is applied? Has this control been done, if yes, it needs to be prominently featured in the discussion.*

Our response:

Thanks for the reviewer's comment. To demonstrate that, we immersed 1 g of α -spodumene into 0.5 M H₂SO₄ + 0.5 wt.% H₂O₂ solution, which is the same solution we used for electrochemical leaching. The particles are soaked for 24 hours. The ICE-AES results show that

negligible (0.05 wt.%) of lithium was leached in this solution. To clarify this, we updated Table S5, and added some discussion into the manuscript.

Change to the manuscript:

We modified the main context and Table S5 with the new ICP-AES results. It measured the 1 g spodumene immersed in 0.5 M H₂SO₄ + 0.5 wt.% H₂O₂ solution for 24 hours. The ICP-AES results show that ~ 0 % of the lithium was leached out.

Without the applied voltage, no lithium is leached out in the same solution for 24 hours. (Page 14, line 15, main context)

Table S5 is updated with the holding test without any applied potential.

Supplementary Table 5. Chemical composition of the leachant by ICP-AES and the leaching efficiencies.

Leaching potential (V vs. SCE)	Feedstock (g)	Theoretical concentration (ppm)	Concentration by ICP (ppm)	Leaching efficiency (%)
No voltage (24 h)	1.0	671	0.35	0.05
0.9	1.0	671	322	48.08
0.9	1.0	671	310	46.20
0.9	1.0	671	314	46.79
0.95	1.0	671	618	92.12
0.95	1.0	671	614	91.51
0.95	1.0	671	607	90.46
1.0	1.0	671	604	91.22

1.0	1.0	671	609	90.76
1.0	1.0	671	610	90.90

8) *Hydrogen peroxide from commercial sources is often reagent grade and may contain a non-trivial amount of metal ions among other impurities- it is also reactive such that we ask our students to avoid metal needles in its handling. Have the authors excluded the possibility of metal ion introduction through the addition of H₂O₂?*

Our response:

Thanks for pointing out this. It is a very practical issue to consider for industrial applications. However, the majority focus of this paper is on fundamental study. For the chemicals we used, the product specification claims the heavy metals < 5 ppm. We employed a pipette for the precise transfer of H₂O₂, and the subsequent analysis by ICP-AES revealed no impurity metal ions, with the detection limit being 0.02 ppm.

9) *Lastly, I was unable to discern what specific particle/crystal size was being utilized in these studies? Is this size uniform? How is this controlled and how does it affect the leaching process efficiency?*

Our response:

Thanks for bringing up this question. For the spodumene used in this work, we did another particle size analysis via laser diffraction besides SEM characterization (details in the experimental section). The average size is determined as 51.0 μm for α-phase and 31.1 μm for β-phase. The distribution curve is added to Fig. S1. Our industrial partner asked this question and we're going to explore the size effect (leaching kinetics) as the next step plan. This work mainly focuses on the fundamentals and leaching mechanism.

Change to the manuscript:

We updated the main context, experimental section, and Fig. S1 with the particle size distributions measured by laser diffraction.

Fig. S1 exhibits that the median particle size of α -phase is 51.0 μm . The median particle size of β -phase decreases to 31.1 μm after phase transformation. (Page 4, line 12, main context)

The particle size analysis used Mastersizer 3000 (Malvern Panalytical) with 632nm laser and 470 blue LED. The spodumene particles were dispersed with DI water. (Page 19, line 28, Experiment section)

Figure S1 is updated.

Supplementary Fig. 1. a, Optical image of α -phase (left) and β -phase (right) spodumene. Size distributions of b, α -phase, and c, β -phase, measured by laser diffraction.

Thank you for the opportunity to review this work and I look forward to a revision.

Thanks again for your encouraging and constructive feedback! These comments helped us improve the overall quality and clarity of our manuscript a lot.

Reference

- 1 Marcandalli, G., Goyal, A. & Koper, M. T. M. Electrolyte Effects on the Faradaic Efficiency of CO(2) Reduction to CO on a Gold Electrode. *ACS Catal* **11**, 4936-4945 (2021). <https://doi.org/10.1021/acscatal.1c00272>
- 2 Fuller, T. F. & Harb, J. N. *Electrochemical engineering*. (John Wiley & Sons, 2018).

- 3 Landon, P. *et al.* Direct synthesis of hydrogen peroxide from H₂ and O₂ using Pd and Au
catalysts. *Physical Chemistry Chemical Physics* **5**, 1917-1923 (2003).
<https://doi.org/10.1039/B211338B>
- 4 Landon, P., Collier, P. J., Papworth, A. J., Kiely, C. J. & Hutchings, G. J. Direct formation
of hydrogen peroxide from H₂/O₂ using a gold catalyst. *Chemical Communications*, 2058-
2059 (2002). <https://doi.org/10.1039/B205248M>
- 5 Pizzutilo, E. *et al.* Gold–Palladium Bimetallic Catalyst Stability: Consequences for
Hydrogen Peroxide Selectivity. *ACS Catalysis* **7**, 5699-5705 (2017).
<https://doi.org/10.1021/acscatal.7b01447>
- 6 Xu, G. *et al.* The Formation/Decomposition Equilibrium of LiH and its Contribution on
Anode Failure in Practical Lithium Metal Batteries. *Angew Chem Int Ed Engl* **60**, 7770-
7776 (2021). <https://doi.org/10.1002/anie.202013812>
- 7 Fang, C. *et al.* Quantifying inactive lithium in lithium metal batteries. *Nature* **572**, 511-515
(2019). <https://doi.org/10.1038/s41586-019-1481-z>
- 8 Wandt, J., Freiberg, A. T. S., Ogrodnik, A. & Gasteiger, H. A. Singlet oxygen evolution
from layered transition metal oxide cathode materials and its implications for lithium-ion
batteries. *Materials Today* **21**, 825-833 (2018).
<https://doi.org/10.1016/j.mattod.2018.03.037>
- 9 Nomura, Y. *et al.* Dynamic imaging of lithium in solid-state batteries by operando electron
energy-loss spectroscopy with sparse coding. *Nat Commun* **11**, 2824 (2020).
<https://doi.org/10.1038/s41467-020-16622-w>
- 10 Grenier, A. *et al.* Nanostructure Transformation as a Signature of Oxygen Redox in Li-
Rich 3d and 4d Cathodes. *J Am Chem Soc* **143**, 5763-5770 (2021).
<https://doi.org/10.1021/jacs.1c00497>
- 11 House, R. A. *et al.* First-cycle voltage hysteresis in Li-rich 3d cathodes associated with
molecular O₂ trapped in the bulk. *Nature Energy* **5**, 777-785 (2020).
<https://doi.org/10.1038/s41560-020-00697-2>
- 12 Luo, K. *et al.* Charge-compensation in 3d-transition-metal-oxide intercalation cathodes
through the generation of localized electron holes on oxygen. *Nat Chem* **8**, 684-691 (2016).
<https://doi.org/10.1038/nchem.2471>
- 13 Petersen, H. A., Myren, T. H. T., O’Sullivan, S. J. & Luca, O. R. Electrochemical methods
for materials recycling. *Materials Advances* **2**, 1113-1138 (2021).
<https://doi.org/10.1039/d0ma00689k>
- 14 Song, Y. & Zhao, Z. Recovery of lithium from spent lithium-ion batteries using
precipitation and electrodialysis techniques. *Separation and Purification Technology* **206**,
335-342 (2018). <https://doi.org/10.1016/j.seppur.2018.06.022>
- 15 Li, Z., He, L., Zhao, Z. w., Wang, D. & Xu, W. Recovery of Lithium and Manganese from
Scrap LiMn₂O₄ by Slurry Electrolysis. *ACS Sustainable Chemistry & Engineering* **7**,
16738-16746 (2019). <https://doi.org/10.1021/acssuschemeng.9b04127>
- 16 Kundu, T., Rath, S. S., Das, S. K., Parhi, P. K. & Angadi, S. I. Recovery of lithium from
spodumene-bearing pegmatites: A comprehensive review on geological reserves,
beneficiation, and extraction. *Powder Technology* **415** (2023).
<https://doi.org/10.1016/j.powtec.2022.118142>

REVIEWERS' COMMENTS

Reviewer #1 (Remarks to the Author):

I am happy with authors' response. The ms has been largely improved. I have no further questions.

Reviewer #2 (Remarks to the Author):

The authors have adequately and thoroughly addressed my questions.